# EVALUATING AND IMPROVING GENERATION CONSISTENCY OF LARGE LANGUAGE MODELS VIA A DIVIDE-CONQUER-REASONING APPROACH

## ABSTRACT

Evaluating the quality and variability of text generated by Large Language Models (LLMs) poses a significant, yet unresolved research challenge. Traditional evaluation methods, such as ROUGE and BERTScore, which measure token similarity, often fail to capture the holistic semantic equivalence. This results in a low correlation with human judgments and intuition, which is especially problematic in high-stakes applications like healthcare and finance where reliability, safety, and robust decision-making are highly critical. This work proposes an automated framework for evaluating the consistency of LLM-generated texts using a divide-and-conquer strategy. Unlike existing LLM-based evaluators that operate at the paragraph level, our method employs a divide-and-conquer evaluator (`DCE`) that breaks down the comparison between two generated responses into individual sentences, each evaluated based on predefined criteria. To facilitate this approach, we introduce an automatic metric converter (`AMC`) that translates the output from `DCE` into an interpretable numeric score. Beyond the consistency evaluation, we further present a reason-assisted improver (`RAI`) that leverages the analytical reasons with explanations identified by `DCE` to generate new responses aimed at reducing these inconsistencies. Through comprehensive and systematic empirical analysis, we show that our approach outperforms state-of-the-art methods by a large margin (e.g., +19.3% and +24.3% on the SummEval dataset) in evaluating the consistency of LLM generation across multiple benchmarks in semantic, factual, and summarization consistency tasks. Our approach also substantially reduces nearly 90% output inconsistencies, showing promise for effective hallucination mitigation and reduction.

## 1 INTRODUCTION

Large language models (LLMs) such as GPT-4 and PaLM 2 (Yang et al., 2023; Bubeck et al., 2023) have demonstrated impressive performance on a variety of natural language generation (NLG) tasks, including summarization (Tam et al., 2022), open-book question-answering (QA) (Kamalloo et al., 2023), and retrieval-augmented generation (RAG) (Lewis et al., 2020; Liu et al., 2023a). However, conventional evaluation methods, such as BARTScore (Yuan et al., 2021) and BERTScore (Zhang et al., 2020), which rely on *token-level* comparison, are inadequate for accurately and reliably measuring the quality of generated content, particularly in complex scenarios with long paragraphs (Liu et al., 2023b; Hanna & Bojar, 2021). To address this issue, LLM-based evaluators such as G-Eval (Liu et al., 2023b) and GPTScore (Jinlan et al., 2023) have proposed a new framework that evaluates texts via *paragraph-level* comparison. While these evaluators show promise for certain tasks, their scores often fail to achieve high concordance with human judgments of semantic equivalence. Furthermore, as only numeric scores are provided with no explanation, it can be challenging for humans to trust or reason about these scores, particularly when using LLMs that are known to hallucinate (Li et al., 2023; Ji et al., 2023; Rawte et al., 2023).

Assessing the consistency of LLMs is more broadly connected to AI safety and has become a critical step in improving the reliability of these systems by preventing the generation of misinformation and harmful content. Wang et al. (2022) demonstrates that *consistency checking* can significantly enhance the chain of thought reasoning in LLMs. Similarly, Kuhn et al. (2023) leverages semantic

consistency for uncertainty estimation in NLG. Recent studies employ consistency checking to detect hallucinations based on pre-trained LLMs (Manakul et al., 2023) and instruction-tuned LLMs (Mündler et al., 2023). Although these methods exhibit promising results on several specific tasks, including mathematical reasoning and factual assessment, the potential failures (Chen et al., 2023) of self-consistency are often overlooked. This is essentially due to a lack of a generic, automatic, and reliable strategy that assesses the consistency of two responses, let alone remediating such inconsistency after identifying them.

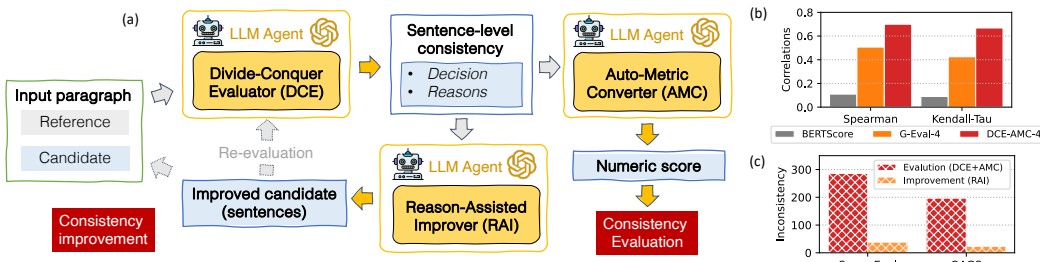

Figure 1: (a) Overview of the proposed `DCR` framework. The first two components (`DCE-AMC`) aim at providing a better strategy for evaluating and quantifying semantic consistency to best match human judgments. Building on this, a third component `RAI` further utilizes analytical reasoning to iteratively improve the consistency of LLM-generated content w.r.t. the reference by minimizing hallucinations. (b) The combination of `DCE` and `AMC` (`DCE-AMC-4`) significantly outperforms the baseline methods in terms of correlations with human ratings. (c) `RAI` substantially reduces output inconsistencies by ∼ 90% through a single improvement iteration on SummEval and QAGS benchmarks.

In this paper, we introduce a novel framework, called `Divide-Conquer-Reasoning` (abbreviated as `DCR` hereafter), for developing an automatic and reliable consistency evaluation method. Our approach capitalizes on the intuition that human evaluators typically assess consistency by comparing the semantic meaning of the generated text to the reference text sentence-by-sentence, and then combining the analysis to make a holistic judgment of the complete concept. Unlike existing metrics that rely on either token-level or paragraph-level checks, our approach is rooted in the sentence level and is better aligned with human judgments. This approach avoids confusing LLM by either providing too much information at once or zooming in too narrowly. Additionally, our approach does not rely on LLMs, which are prone to hallucination, to output numeric scores without justification. Another advantage of our approach is its ability to mitigate inconsistencies after identifying them.

The `DCR` framework is composed of three components, each executed by an LLM agent, as shown in Fig. 1. Given the reference and candidate, the Divide-Conquer Evaluator (`DCE`) realizes the notion of divide-conquer to determine whether the candidate is semantically equivalent to the reference at a sentence level. `DCE` automatically partitions the candidate paragraph into sentences (*divide*), evaluates each sentence against the reference paragraph based on pre-defined semantic-level consistency criteria (*conquer*), and generates a list of reasons that explain why each sentence is or is not consistent with the reference paragraph. Next, the Auto-Metric Converter (`AMC`) which builds upon `DCE`, converts the reasons (with explanations) into a numeric score system that is more intuitive for humans to comprehend and evaluate the performance of `DCE`. The numeric score can be used to evaluate consistency in various tasks, such as summarization, factual assessment, and hallucination detection.

Our `DCR` framework not only evaluates consistency but also enhances it through the Reason-Assisted Improver (`RAI`), a third LLM agent that utilizes the outputs of `DCE` to generate new candidate sentences. By incorporating the explanations provided by `DCE` with the original context, `RAI` produces sentences that mitigate inconsistencies (hallucinations). This improvement process can be iteratively applied by utilizing the re-evaluation produced by `DCE` to ultimately achieve a candidate response that is fully aligned with the reference text.

We conducted an evaluation of our approach on three different NLG tasks, including semantic, summarization, and factual consistency evaluations. Our results demonstrate that `DCR` significantly outperforms all existing baseline methods as a consistency evaluator, with improvements of up to 19.3% and 24.3% compared to G-Eval on the SummEval dataset. Additionally, our approach achieved high correlations with human judgment on all three benchmarks. Notably, we observed highly promising results in consistency improvement rate (from 86.71% to 91.11%) at a substantially lower effort and cost due to its multi-thread parallel implementation.

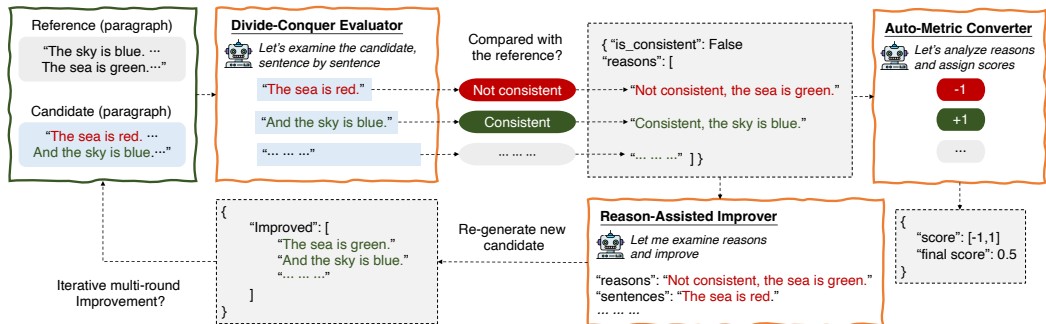

Figure 2: An example of evaluating and improving the consistency of generated text via `DCR`.

## 2 PRELIMINARIES

**Black-Box LLM Evaluation.** One of the drawbacks of current grey-box LLM evaluations is that they require output token-level probabilities (Jiang et al., 2023). However, prominent LLMs such as GPT-3.5, GPT-4, PaLM 2, and Claude 2, are only available through restricted API calls. Therefore, such token-level information might not be available. By contrast, in this paper, we focus on the design of a black-box approach that remains applicable even when only text-based responses are available from the LLM; that is, we only have access to the model output.

**Problem Formulation.** Given a user query $\mathcal{Q}$ and LLM model $\mathcal{M}$, let $\mathcal{C}$ refer to the candidate response drawn from $\mathcal{C} = \mathcal{M}(\mathcal{Q})$. LLM-generated responses are commonly evaluated using some reference texts, denoted by $\mathcal{R}$, for instance, human writing samples for generation tasks and original content for summarization tasks. The objective of consistency evaluation is to build a function $f$ that quantitatively measures the *semantic equivalence* $\mathcal{S}$ between the generated candidates $\mathcal{C}$ and reference $\mathcal{R}$ as $\mathcal{S} = f(\mathcal{R}, \mathcal{C}|\mathcal{Q}, \mathcal{M})$ where $\mathcal{S}$ could be binary decision, such as "Yes" or "No", "Consistent" or "Not Consistent", or numeric score, e.g., [-1, +1]. However, it is worth noting that our evaluation can be generally used to check consistency between two candidates where both are generated by LLMs. In that scenario, we only need to assume one candidate as the reference for self-check consistency.

**Limitation of Existing Methods.** The conventional metrics, such as BERTscore and BARTscore, rely on a *token-level* comparison using n-gram or contextual embedding to calculate cosine-similarity. However, this approach fails to capture the overall semantic meaning as it directly aggregates token-level similarities. To address this issue, leveraging the power of LLMs for self-evaluation has been proposed. G-Eval (Liu et al., 2023b) and GPT-Eval (Jiang et al., 2023) evaluate consistency at a paragraph-level by prompting LLMs to compare two candidates as a whole. However, these approaches have a major drawback as the generated verbal scores by LLMs are *prone to hallucinations*, resulting in abnormally higher ratings for LLM-generated content that diverge from human judgment (Liu et al., 2023b). Such methods also generate no actionable insight to justify the score or mitigate inconsistencies after identifying them.

## 3 DIVIDE-CONQUER-REASONING

To overcome the aforementioned limitations, we propose to evaluate and improve the consistency of LLM output via a Divide-Conquer-Reasoning approach, referred to as `DCR`. The approach comprises three key components, as illustrated in Fig. 1: (1) `DCE`, which disassembles the candidate paragraph and scrutinizes semantic inconsistencies sentence-by-sentence, (2) `AMC`, which converts sentence-level inconsistency/consistency reasons into numeric scores for quantitative interpretation, and (3) `RAI`, which conducts analytical reasoning to improve consistency through candidate regeneration. Essentially, our approach involves a combination of sentence-level analysis, semantic consistency checking, and causal analysis, making it an ideal evaluation metric for a diverse range of NLG tasks that require comparison to reference texts, such as summarization, open-book question-answering (QA), and retrieval-augmented generation.

Moreover, `DCR` not only evaluates but also improves the consistency of generated text through analysis and reasoning, which aligns with human intuition. Fig.2 provides an example of how `DCR` can evaluate and enhance the consistency of candidate text. In the following sections, we will discuss each component in detail.

### 3.1 DIVIDE-CONQUER EVALUATOR (DCE)

The Divide-Conquer Evaluator (DCE) is an LLM Agent designed to perform semantic consistency checks between the reference and the candidate using a sentence-by-sentence strategy. This agent accepts a reference paragraph and a candidate paragraph as inputs, and employs a divide-conquer strategy to break down the entire paragraph into multiple individual sentences (*divide*) and then assess each sentence against the reference (*conquer*). More specifically, given the input reference $\mathcal{R} = \langle s_1^r, ..., s_l^r \rangle$ and candidate $\mathcal{C} = \langle s_1^c, ..., s_k^c \rangle$, we build a DCE agent $\mathcal{L}_{\text{DCE}}$ using the LLM model $\mathcal{M}$ (e.g., GPT-3.5/4) with an instructed prompt $\mathcal{P}_{\text{DCE}}$ as:

$$\{\gamma_1, \gamma_2, ..., \gamma_k\} = \mathcal{L}_{\text{DCE}}(\langle s_1^c, s_2^c, ..., s_k^c \rangle, \mathcal{R} \mid \mathcal{M}, \mathcal{P}_{\text{DCE}}). \tag{1}$$

Eq.1 generates *reasons*, denoted as $\Gamma = \{\gamma_1, \gamma_2, ..., \gamma_k\}$, which is a list of reasons explaining why each sentence $s_i^c(i = 1, 2, ..., k)$ is or is not consistent against the *entire* reference paragraph $\mathcal{R}$. It's important to note that the reasons $\gamma_i$ might comprise a short paragraph containing multiple explanation sentences. We can tailor instruction prompts by defining task-specific criteria to accommodate different tasks. Table 1 provides an example of a prompt example with pre-defined criteria for the summarization consistency task.

Table 1: Summarization Consistency Divide-Conquer Evaluator Prompt

---

Your task is to evaluate whether the summary is consistent with the article. You will evaluate it by going through each sentence of the summary and check against the following procedures:

- *Understands all the aspects of the sentence, and compare if each aspect exists in the article*
- *If it does, compare if the information in this sentence is consistent with what is in the article*
- *Compare if all the information in this sentence can be directly inferred or entailed from what is in the article. It is OK that not all information from the article exists in this summary*

---

### 3.2 AUTO-METRIC CONVERTER (AMC)

The Auto-Metric Converter (AMC) is an LLM Agent that aims to quantitatively measure the consistency evaluation derived from the Divide-Conquer Evaluator (DCE) by converting the reasons from DCE into a numeric score system. This is accomplished by introducing an LLM agent, denoted as $\mathcal{L}_{\text{AMC}}$, which takes reasons $\langle \gamma_1, \gamma_2, ..., \gamma_k \rangle$ with an instructed prompt $\mathcal{P}_{\text{AMC}}$ as inputs:

$$\{z_1, z_2, ..., z_k\} = \mathcal{L}_{\text{AMC}}(\{\gamma_1, \gamma_2, ..., \gamma_k\} \mid \mathcal{M}, \mathcal{P}_{\text{AMC}}). \tag{2}$$

The LLM Agent $\mathcal{L}_{\text{AMC}}$ functions as a binary sentiment classifier that classifies the reasons $\langle \gamma_1, \gamma_2, ..., \gamma_k \rangle$ to be either positive (marked by "+1" if the sentence is consistent), or negative (marked by "-1" otherwise). As a result, AMC outputs an array of scores $\{z_1, z_2, ..., z_k\}, z_i \in \{-1, +1\}$ for each sentence $\langle s_1^c, s_2^c, ..., s_k^c \rangle$ in the candidate $\mathcal{C}$. We then utilize this score array to calculate a comprehensive score $\mathcal{Z}$ to evaluate how consistent the candidate (paragraph) is against the reference (paragraph):

$$\mathcal{Z} = \left( \sum_{i=1}^k z_i + \alpha \right) / (k + \beta), \quad \hat{\mathcal{Z}} = (\mathcal{Z} + 1)/2, \quad \hat{\mathcal{Z}} \in [0, 1] \tag{3}$$

where $k$ is the length of the score array, i.e., the number of sentences in the candidate paragraph. Depending on the prompt, the *reasons* output by DCE may not all be on the sentence level. To ensure that the score calculated is solely generated by sentence-level *reasons*, we introduce $\alpha$ and $\beta$ in Eq. 3, as explained in detail in Appendix A.4. Finally, we rescale $\mathcal{Z}$ to obtain the final score $\hat{\mathcal{Z}}$ that is typically between 0 (*completely inconsistent*) and 1 (*completely consistent*). The closer this score $\hat{\mathcal{Z}}$ is to 0, the more inconsistent the candidate $\mathcal{C}$ is against the reference $\mathcal{R}$.

### 3.3 REASON-ASSISTED IMPROVER (RAI)

The Reason-Assisted Improver (RAI) is an LLM Agent that focuses on improving the consistency of candidate sentences by reasoning through the inconsistent explanations generated by the Divide-Conquer Evaluator (DCE). To achieve this goal, we propose an LLM agent $\mathcal{L}_{\text{RAI}}$ to generate new

candidate sentences $\langle \hat{s}_1^c, \hat{s}_2^c, ..., \hat{s}_k^c \rangle$ based on the collected reasons $\{\gamma_1, \gamma_2, ..., \gamma_k\}$ and original sentences $\langle s_1^c, s_2^c, ..., s_k^c \rangle$:

$$\langle \hat{s}_1^c, \hat{s}_2^c, ..., \hat{s}_k^c \rangle = \mathcal{L}_{\mathtt{RAI}}(\{\gamma_1, \gamma_2, ..., \gamma_k\}, \langle s_1^c, s_2^c, ..., s_k^c \rangle, \mathcal{R} \mid \mathcal{M}, \mathcal{P}_{\mathtt{RAI}}). \tag{4}$$

The core task of $\mathcal{L}_{\mathtt{RAI}}$ is to rewrite the original sentence $s_i^c$ if $s_i^c$ is inconsistent with the reference $\mathcal{R}$ and return a new generated $\hat{s}_i^c$ ($\hat{s}_i^c \neq s_i^c$), otherwise retain $s_i^c$. The newly generated responses $\hat{\mathcal{C}} = \langle \hat{s}_1^c, \hat{s}_2^c, ..., \hat{s}_k^c \rangle$ can be considered as the consistency-improved candidate, which can be re-evaluated by $\mathtt{DCE}$ to check if $\hat{\mathcal{C}}$ mitigates inconsistencies against the reference $\mathcal{R}$.

The improved candidate $\hat{\mathcal{C}}$ in Eq.4 can be directly fed to the $\mathtt{DCE}$ agent in Eq.1 after the *first-round* $\mathtt{DCR}$, i.e., $\mathtt{DCE} \rightarrow \mathtt{AMC} \rightarrow \mathtt{RAI}$. A straightforward extension is *multi-round* consistency improvement, where the consistency is iteratively improved until reaching the maximum number of rounds $m$. Algorithm 1 illustrates the workflow of the $\mathtt{DCR}$ framework, which consists of three core components: $\mathtt{DCE}$, $\mathtt{AMC}$, and $\mathtt{RAI}$.

---

**Algorithm 1** Proposed `Divide-Conquer-Reasoning` (DCR) framework

---

1: **Requirements**: Candidate $\mathcal{C}$, Reference $\mathcal{R}$, LLM model $\mathcal{M}$, LLM agents $\mathcal{L}_{\mathtt{DCE}}, \mathcal{L}_{\mathtt{AMC}}, \mathcal{L}_{\mathtt{RAI}}$ with instructed prompts $\mathcal{P}_{\mathtt{DCE}}, \mathcal{P}_{\mathtt{AMC}}$ and $\mathcal{P}_{\mathtt{RAI}}$, and the maximum number of rounds $m$
1: **for** rounds $r = 1, ..., m$ **do**
2:     Disassemble candidate $\mathcal{C}$ into sentences $\langle s_1^c, s_2^c, ..., s_k^c \rangle$, evaluate sentence-level consistency against reference $\mathcal{R}$, and return the reasons $\{\gamma_1, \gamma_2, ..., \gamma_k\} \leftarrow \mathcal{L}_{\mathtt{DCE}}(\langle s_1^c, s_2^c, ..., s_k^c \rangle, \mathcal{R} \mid \mathcal{M}, \mathcal{P}_{\mathtt{DCE}})$ in Eq. 1
3:     Transform reasons into numeric scores $\{z_1, z_2, ..., z_k\} \leftarrow \mathcal{L}_{\mathtt{AMC}}(\{\gamma_1, \gamma_2, ..., \gamma_k\} \mid \mathcal{M}, \mathcal{P}_{\mathtt{AMC}})$ in Eq. 2
4:     Calculate the final consistency evaluation score $\hat{\mathcal{Z}}$ based on $\{z_1, z_2, ..., z_k\}$ using Eq. 3
5:     Generate improved candidate $\langle \hat{s}_1^c, \hat{s}_2^c, ..., \hat{s}_k^c \rangle \leftarrow \mathcal{L}_{\mathtt{RAI}}(\{\gamma_1, \gamma_2, ..., \gamma_k\}, \langle s_1^c, s_2^c, ..., s_k^c \rangle, \mathcal{R} \mid \mathcal{M}, \mathcal{P}_{\mathtt{RAI}})$
6:     Update the candidate $\langle s_1^c, s_2^c, ..., s_k^c \rangle \leftarrow \langle \hat{s}_1^c, \hat{s}_2^c, ..., \hat{s}_k^c \rangle$ and return Step 2
7: **return** $\hat{\mathcal{Z}}, \langle \hat{s}_1^c, \hat{s}_2^c, ..., \hat{s}_k^c \rangle$

---

## 4 EXPERIMENTS

### 4.1 BENCHMARKS AND IMPLEMENTATION DETAILS

We utilize GPT-3.5 (`gpt-3.5-turbo`) and GPT-4 (`gpt-4`) as our LLM agents, and the evaluations are carried out using the Azure OpenAI API. We set the temperature to 0.0 to generate responses via the greedy algorithm. The specific prompts used for each LLM agent are detailed in the Appendix (from Table 7 to Table 12). All experiments are conducted on our local machine (Macbook-Pro with M1 chip) without the need for GPU resources. In our experimental setup, we set both $\alpha$ and $\beta$ in Eq. 3 to 0. We employ four datasets to evaluate $\mathtt{DCR}$ where QQP and PAWS are binary datasets, as well as SummEval and QAGS have numeric scores representing human judgments.

- **QQP and PAWS**: Quora Question Pair corpus (Iyer et al., 2017) and the Paraphrase Adversaries from Word Scrambling dataset (amd Jason Baldridge & He, 2019) contain pairs of sentences labeled to indicate whether they are paraphrases or not, while PAWS specifically focuses on the adversarial paraphrases. Following the guidance of BERTScore (Zhang et al., 2020), we are using the PAWSQQP development set and the first 5000 from the training set of QQP.

- **SummEval** (Fabbri et al., 2021) is a standard dataset that assesses various summarization evaluation techniques. It gathers human ratings in various aspects and is built on the CNN/DailyMail dataset (Hermann et al., 2015). In this study, we mainly focus on the consistency evaluation.

- **QAGS** (Wang et al., 2020) serves as a benchmark for assessing hallucinations in summarization tasks. Its objective is to evaluate the consistency aspect of summaries across two distinct summarization datasets: QGS-CNN and QAGA-XSUM.

### 4.2 BASELINES

We evaluate $\mathtt{DCR}$ against a variety of evaluation metrics and LLM-based evaluators that have achieved state-of-the-art performance.

- **ROUGE** (Lin, 2004) is widely used evaluation metric with three different variants ROUGE-1, ROUGE-2, and ROUGE-L. We are using ROUGE-2 and ROUGE-L as comparisons in our study.
- **BERTScore** (Zhang et al., 2020) calculates the similarities between two pieces of text using the contextualized embedding derived from the BERT model(Devlin et al., 2019). It operates as a similarity-based assessment tool, which has been widely used for various applications.

- **MoverScore** (Zhao et al., 2019) enhances BERTScore by incorporating soft alignments and introducing new aggregation techniques to provide a more robust similarity assessment.
- **BARTScore** (Yuan et al., 2021) is a comprehensive evaluator that uses the average likelihood of the model's output as its measurement criteria.
- **UniEval** (Zhong et al., 2022) is a consolidated evaluator capable of assessing various elements of text generation as QA tasks. It manages diverse evaluation tasks by modifying the question format.
- **GPTScore** (Jinlan et al., 2023) is an LLM-based evaluator that assesses texts using pre-training models, e.g., GPT-3, and is designed to provide a higher likelihood to high-quality generated text.
- **G-Eval** (Liu et al., 2023b) is another LLM evaluator that utilizes LLMs with a chain-of-thoughts (CoT) approach with a form-filling paradigm to evaluate the quality of NLG outputs.

### 4.3 MAIN RESULTS ON CONSISTENCY EVALUATION (DCE-AMC)

**Semantic Consistency Evaluation.** Table 2 shows the Area Under the ROC curve (AUROC) for automatic baseline metrics and our method, following the practice of BERTScore (Zhang et al., 2020). We note that while most metrics from BERTScore perform acceptably on QQP, they exhibit a significant performance drop on $PAWS_{QQP}$. This suggests that these baseline metrics struggle to detect the challenging adversarial examples from a semantic consistency perspective. In contrast, our method, whether implemented with GPT-3.5 or GPT-4, outperforms all the baseline metrics on both QQP and $PAWS_{QQP}$, without a significant drop. Notably, DCE-AMC-4 demonstrates superior robustness in adversarial paraphrase classification (semantic consistency) achieving a relatively large improvement (+4.6% in QQP and +9.9% in $PAWS_{QQP}$) compared to BERTScore.

Table 2: AUROC results on QQP and $PAWS_{QQP}$

| Metrics | QQP | $PAWS_{QQP}$ |
|---|---|---|
| BLEU | 0.707 | 0.527 |
| METEOR | 0.755 | 0.532 |
| ROUGE-L | 0.740 | 0.536 |
| CHRF++ | 0.577 | 0.608 |
| BEER | 0.741 | 0.564 |
| EED | 0.743 | 0.611 |
| CharacTER | 0.698 | 0.650 |
| BERTScore | 0.777 | 0.693 |
| DCE-AMC-3.5 | 0.788 | 0.770 |
| DCE-AMC-4 | **0.823** | **0.792** |

**Factual Consistency Evaluation.** While advanced NLG models are capable of generating high-quality responses, LLMs are known to occasionally produce non-factual statements or hallucinat facts, which can undermine trust in their output. Recent work (Manakul et al., 2023) has been conducted to identify such inconsistencies in terms of factuality. To verify the effectiveness of our method in evaluating hallucination, we test it on the QAGS benchmark, which includes two summarization datasets: QAGS-CNN and QAGS-XSUM. Table 4 provides a comprehensive comparison of various metrics based on Pearson, Spearman, and Kendall-Tau correlations. We observe that BARTScore performs competitively on the extractive subset (QAGS-CNN) but fails to demonstrate a high correlation on the abstractive subset (QAGS-XSUM). UniEval exhibits a better correlation than G-Eval-3.5 but is comparable to G-Eval-4. Our proposed DCE-AMC-4 outperforms all the baseline methods on both subsets, particularly by a significant margin on QAGS-XSUM. Unlike the G-Eval method, which shows a larger gap between GPT-3.5 and GPT-4, our DCE-AMC method remains relatively stable when switching between LLMs.

It's crucial to note that QAGS-XSUM is an abstractive dataset, and its summaries are typically one sentence long. This contrasts with the extractive database of QAGS-CNN, where summaries are composed of multiple sentences. Consequently, our method operates at a sentence level for QAGS-XSUM, and our final score is always either 0 or 1. Furthermore, the binary label in QAGS-XSUM implies that we achieve the same correlation score using different correlation methods.

**Summarization Consistency Evaluation.** We follow the setting of previous work (Zhong et al., 2022) to evaluate different summarization consistency using summary-level Spearman ($\rho$) and Kendall-Tau ($\tau$) correlation. As shown in Table 3, baseline metrics using semantic similarity, such as ROUGE and BERTScore, perform poorly on consistency evaluations. While LLM-based evaluators like GPT-Score and G-Eval exhibit higher correlations, they still underperformed compared to

Table 3: Correlation ($\rho$ and $\tau$) results of different metrics on SummEval benchmark.

| Metrics | SummEval-Consistency | |
|---|---|---|
| | Spearman ($\rho$) | Kendall-Tau ($\tau$) |
| ROUGE-2 | 0.187 | 0.155 |
| ROUGE-L | 0.115 | 0.092 |
| BARTScore | 0.382 | 0.315 |
| BERTScore | 0.110 | 0.090 |
| MoverScore | 0.152 | 0.127 |
| UniEval | 0.446 | 0.371 |
| GPT-Score | 0.449 | - |
| G-Eval-3.5 | 0.386 | 0.318 |
| G-Eval-4 | 0.507 | 0.425 |
| DCE-AMC-3.5 | 0.592 | 0.563 |
| DCE-AMC-4 | **0.700** (+19.3%) | **0.668** (+24.3%) |

our proposed method. `DCE-AMC-4` achieves much higher human correspondence compared to `DCE-AMC-3.5` on both Spearman and Kendall-Tau correlation, which indicates that the larger size of GPT-4 model is beneficial for sentence-level consistency checking in summarization tasks. `DCE-AMC-4` with stronger correlations of $\rho = 0.7$ and $\tau = 0.668$, substantially improves upon the G-Eval-4 baseline by a large margin (+19.3% and +24.3% respectively).

Table 4: Pearson ($r$), Spearman ($\rho$), and Kendall-Tau ($\tau$) correlations of different baseline metrics on QAGS-CNN and QAGS-XSUM benchmark.

| Metrics | QAGS-CNN | | | QAGS-XSUM | | |
|---|---|---|---|---|---|---|
| | Pearson ($r$) | Spearman ($\rho$) | Kendall-Tau ($\tau$) | Pearson ($r$) | Spearman ($\rho$) | Kendall-Tau ($\tau$) |
| ROUGE-2 | 0.459 | 0.418 | 0.333 | 0.097 | 0.083 | 0.068 |
| ROUGE-L | 0.357 | 0.324 | 0.254 | 0.024 | -0.011 | -0.009 |
| BARTScore | 0.735 | 0.680 | 0.557 | 0.184 | 0.159 | 0.130 |
| BERTScore | 0.576 | 0.505 | 0.399 | 0.024 | 0.008 | 0.006 |
| MoverScore | 0.414 | 0.347 | 0.271 | 0.054 | 0.044 | 0.036 |
| UniEval | 0.682 | 0.662 | 0.532 | 0.461 | 0.488 | 0.399 |
| G-Eval-3.5 | 0.477 | 0.516 | 0.410 | 0.211 | 0.406 | 0.343 |
| G-Eval-4 | 0.631 | 0.685 | 0.591 | 0.558 | 0.537 | 0.472 |
| DCE-AMC-3.5 | 0.699 | 0.648 | 0.596 | 0.573 | 0.573 | 0.573 |
| DCE-AMC-4 | **0.782** | **0.760** | **0.706** | **0.602** | **0.602** | **0.602** |

### 4.4 RESULTS FOR CONSISTENCY IMPROVEMENT (`RAI`)

After implementing `DCE` and `AMC`, we can quantitatively determine whether each candidate is consistent (score = 1) to the reference or not (score <1). Table 5 offers a statistical analysis of the number of inconsistent data after evaluations (`DCE-AMC`), revealing 286, 111, and 86 inconsistent candidates for the SummEval, QAGS-CNN, and QAGS-XSUM benchmarks respectively. Identifying these inconsistent candidates is valuable but the more critical objective is how to improve these responses to align with the references. To achieve this goal, we re-generate a new response by implementing `RAI` based on the reasons provided by `DCE`, and then use `DCE` to re-evaluate these improved responses. We observe a significant improvement, with most inconsistencies corrected, specifically 84 out of 86 examples on the QAGS-XSUM benchmark. The rate of consistency improvement is 86.71%, 88.29%, and 97.67% on SummEval, QAGS-CNN and QAGS-XSUM respectively. These impressive results demonstrate that our reasoning approach `RAI` not only provides better consistency evaluation metrics that align more closely with human judgments, but also sheds light on improving consistency beyond evaluation. This finding is particularly crucial for mitigating hallucination once we detect non-factual statements via consistency checks. It's worth noting that our reasoning method `RAI` is a generic component that can also be applied directly at the paragraph level, and the improvement in this context is significant as well, as illustrated in Table 5.

Table 5: Performance of consistency improvement with `RAI` on three benchmark datasets.

| Dataset (size) | SummEval (1600) | | QAGS-CNN (236) | | QAGS-XSUM (239) | |
|---|---|---|---|---|---|---|
| | Sentence | Paragraph | Sentence | Paragraph | Sentence | Paragraph |
| Inconsistent data | 286 | 209 | 111 | 68 | 86 | 90 |
| Corrected data with `RAI` | 248 | 198 | 89 | 64 | 84 | 82 |
| Consistency improvement rate | 86.71% | 94.73% | 88.29% | 94.11% | 97.67% | 91.11% |

### 4.5 ANALYSIS

**Why `DCR` Prefers Sentence-level Evaluation?**    To further assess the potential advantage of the sentence-level approach in consistency checking, we employed the same logic of outputting decisions and reasons as used in `DCE` and developed an evaluator at the paragraph level, with prompts provided in Appendix (Table 11). The comparative results between paragraph level and sentence level can be viewed in Fig. 3. While the recall of paragraph-level evaluation is higher on SummEval and QAGS-CNN benchmarks, its overall performance in terms of the F1 score and precision is lower than that of sentence-level evaluations, particularly on the QAGS benchmark. This combination of higher recall and lower precision implies that more candidates are incorrectly marked as consistent. In the task of consistency checking, a metric with low recall and high precision (sentence level) is

preferable as it contributes to higher safety compared to a metric with high recall and low precision (paragraph level), erring on the side of caution.

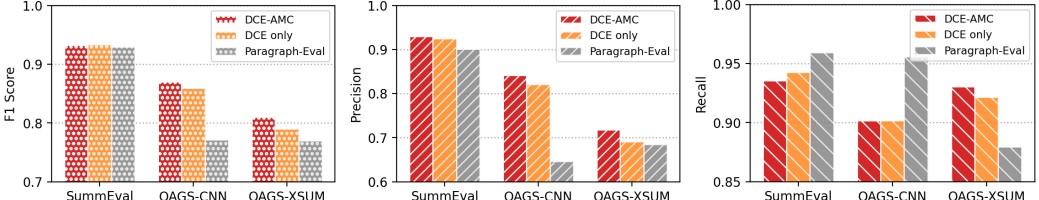

Figure 3: F1 score, precision, and recall performance of our method on sentence-level and paragraph-level evaluations. We also verify the effectiveness of the auto-metric converter.

In addition to superior accuracy, sentence-level evaluations can facilitate more thorough inconsistency remediation when integrating with `RAI`. We compared the performance improvement between our sentence level `DCE` and paragraph level, as indicated in Table 5. Despite the higher recall of the paragraph-level approach, fewer items are flagged as inconsistent, resulting in fewer candidates being corrected, even though the improvement rate is higher. In fact sentence level `DCE` leads to 25.25% and 39.05% more corrections compared to the paragraph-level approach in SummEval and QAGS-CNN respectively. Therefore, our sentence-level approach not only outperforms in terms of F1 score and precision during consistency checks, but also facilitates comprehensive improvements through `RAI`.

**Is Auto-metric Converter Necessary?**   We present a comparison of our method, both with and without `AMC`, as shown in Fig. 3. We observe that our method with only the `DCE` *(red bar)* performs marginally better on the SummEval dataset but underperforms `DCE-AMC` *(orange bar)* on all other benchmarks. Although `DCE` plays a key role in our method, the `AMC` component is still desirable and highly necessary not only because it shows better performance, but also because it facilitates the conversion of *reasons* outputted by `DCE` to a numeric system. This conversion is both user-friendly and practical, making it easy for humans to understand and apply. Furthermore, it provides a straightforward means of evaluating the effectiveness of the `DCE` component.

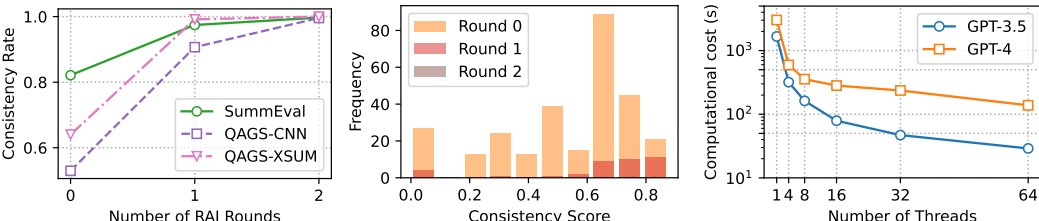

Figure 4: Multi-round consistency improvement          Figure 5: Computational cost.

**Multi-round Consistency Improvement.**   Table 5 showcases encouraging results on consistency improvement via `RAI`. This naturally leads to the question: can we further enhance the consistency through multiple rounds of `RAI`? Fig. 4 shows our investigation on multi-round consistency improvement by iteratively applying `RAI`. It is noteworthy that the convergence of consistency improvement is remarkably swift, achieving nearly 100% in just two rounds. The convergence rate on the QAGS datasets is highly consistent across both subsets, slightly surpassing SummEval due to its high initial rate after the first round of `RAI`. This is also corroborated by the frequency distribution of the consistency score (Fig. 4 (right)). As the number of rounds increases, the lower consistency scores (<1) gradually decrease, and more inconsistent candidates tend to be consistent, where the score is 1.

**The Effect of LLM models.**   We evaluated the performance of our method using different LLMs across all three benchmarks. It is noteworthy that `DCE-AMC`-4 generally outperforms `DCE-AMC`-3.5 across most datasets. The performance gap between the two LLM models is relatively minor in terms of semantic consistency (QQP and PAWS$_{QQP}$ in Table 2), and the abstractive subset (QAGS-XSUM in Table 4) in factual consistency, but a significant difference is observed in summarization consistency in Table 3. This suggests that GPT-4 can further enhance performance, especially for more complex evaluation tasks. As such, we applied `RAI` with GPT-4 directly to verify its superior capability in consistency improvement. Nonetheless, the benefits of GPT-3.5, such as higher computational efficiency and lower API costs, should not be overlooked.

**Computational Cost.**   We assessed the computational cost of our method based on wall-clock time, which is primarily consumed by LLMs inference. However, the divide-conquer strategy we employed

is scalable and easily implemented in parallel. Fig. 5 illustrates the computational cost of GPT-3.5 and GPT-4 with varying numbers of threads on the QAGS-CNN benchmark. A clear reduction in computational cost is observed as the number of threads increases. It's important to note that the decrease in time is more significant when transitioning from a single thread to four threads, but tends to plateau as more threads are utilized. While GPT-3.5, being the smaller LLM, is a more efficient option, GPT-4 often delivers better performance.

## 5 RELATED WORK

**LLM-based Evaluations.** Unlike conventional evaluating metrics leveraging token-level or similarity embeddings, such as BLEU (Papineni et al., 2002), ROUGE (Lin, 2004), or BERTScore (Zhang et al., 2020), recent proposed LLM-based evaluators (Wang et al., 2023), such as GPTScore (Jinlan et al., 2023) and G-Eval (Liu et al., 2023b), have demonstrated competitive performance on multiple NLG tasks. Their idea is to utilize the LLMs to score the candidate output under the assumption that the LLMs have learned to assign higher probabilities to fluent and high-quality contexts. However, these LLM evaluators often exhibit lower correlations with human judgments, and their reliability, robustness, and validity remain under-explored (Liu et al., 2023b). Specifically, LLM evaluators may pose potential risks of producing hallucinated or overconfidence scores if the LLM model is not well calibrated for complex tasks (Kadavath et al., 2022; Zhou et al., 2023). This results in limited confidence in using LLM evaluators to directly evaluate paragraph-level responses. Our proposed DCR framework addresses these challenges through a divide-conquer strategy (DCE) coupled with a numeric score system (AMC). Our method quantitatively evaluates paragraphs sentence-by-sentence and does not rely on LLMs to directly output numeric scores, thus providing a more accurate and comprehensive score that better aligns with human feedback.

**Consistency Evaluations.** Consistency checking plays an essential role in a wide range of NLG tasks, including question-answering (Durmus et al., 2020; Wang et al., 2020), factual knowledge extraction (Elazar et al., 2021), summarization (Durmus et al., 2020) and hallucination detection (Manakul et al., 2023). However, due to various limitations of existing methods, such as reliance on additional pre-trained models or question sets (Durmus et al., 2020), it is highly desirable to develop a unified and automatic consistency metric (Wang et al., 2022). Our proposed framework successfully fills this gap and demonstrates superior performance compared to state-of-the-art baselines (Jinlan et al., 2023; Liu et al., 2023b; Wang et al., 2023). More importantly, our proposed RAI enables consistency improvement where the re-generated candidate response significantly helps mitigate LLM hallucinations (Dhuliawala et al., 2023; Mündler et al., 2023; Zhang et al., 2023) in summarization, and open-book QA tasks (Li et al., 2023).

## 6 CONCLUSION AND DISCUSSION

We proposed a general evaluation framework based on a divide-and-conquer strategy for assessing the consistency between the LLM-generated output and the reference texts across various NLG tasks. Moreover, the proposed method can leverage analytical reasoning to generate revised text with improved consistency. Through comprehensive and systematic empirical study across multiple benchmarks in semantic, factual, and summarization consistency tasks, we demonstrated that our approach significantly outperforms existing methods in evaluating and enhancing the consistency of LLM-generated content. Despite these advancements, we acknowledge several potential limitations of our proposed method:

**Not a Silver Bullet.** While our sentence-level approach (DCE-AMC) excels in evaluating *consistency* and *detecting hallucination*, it may not be universally effective for all dimensions of text evaluation, even with updated criteria in prompts. For instance, dimensions such as *coherence*, which pertains to the collective quality of all generated sentences, or *relevance*, which involves selecting important information and eliminating redundant content from the reference text, require a holistic focus on the entire candidate. These dimensions may not be ideally suited for our DCE-AMC approach. However, if a different evaluator that outputs reasons for action is used, our AMC and RAI could still be employed to quantify and improve performance on such dimensions.

**Garbage in, Garbage Out.** The DCR framework requires two inputs: a reference paragraph and a candidate paragraph. As we use the reference paragraph as the target for consistency and hallucination checks, any non-factual statements present in the reference paragraph would not be detected by our method. Therefore, for tasks such as retrieval-augmented generation (RAG), the accuracy of our method is inherently limited by the correctness of the input paragraphs.

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

# A APPENDIX

## A.1 IMPROVEMENT ITERATIONS EXAMPLE

Table 6: SummEval Improvement Iteration Example

| | |
|---|---|
| **Original Article** | Usain Bolt will compete at the IAAF/BTC World Relays in the Bahamas next month , the Jamaica Athletics Administrative Association has announced . The six-time Olympic gold medallist will compete at the relay championship on May 2 and 3 as part of the Jamaican team . ' I 'm happy to be part of the Jamaican team for the IAAF / BTC World Relays in the Bahamas . I am fit , healthy and ready to run , ' said Bolt . Usain Bolt has confirmed he will be part of Jamaica 's team at the World Relays in the Bahamas Bolt reacts as he wins 4x100m gold at the London Olympic Games in 2012 ' I hear the meet was a lot of fun last year and there was a great atmosphere . Jamaica has a long and successful tradition in relays and when we put on the national colours we always do our best to make the country proud , ' he added . JAAA General Secretary Garth Gayle commented , 'We were extremely pleased that Usain was available for selection and that the world 's fastest man will be running for Jamaica . We can expect some sprint magic on the track in the Bahamas on 2nd and 3rd May . ' ... |
| **Original Summary** | Usain bolt will compete at the relay championship on may 2 and 3 as part of the jamaican team . The six-time olympic gold medalist will be part of jamaica 's team at the iaaf/btc world . Bolt is the iaaf/btc general secretary garth gayle said he always does ' best to make his country proud ' . ... |
| **Divide-Conquer Evaluator Evaluation Reasons** | • This sentence is consistent with the article. The article states that Usain Bolt will compete at the IAAF/BTC World Relays in the Bahamas next month as part of the Jamaican team
• This sentence is consistent with the article. The article confirms that Usain Bolt, a six-time Olympic gold medallist, will be part of Jamaica's team at the IAAF/BTC World Relays
• This sentence is not consistent with the article. The article does not state that Bolt is the IAAF/BTC General Secretary. The General Secretary is Garth Gayle. However, the article does mention that Bolt always does his best to make his country proud

... |
| **First Round Improved Answer** | Usain bolt will compete at the relay championship on may 2 and 3 as part of the jamaican team. The six-time olympic gold medalist will be part of jamaica 's team at the iaaf/btc world. Bolt, as stated by the IAAF/BTC General Secretary Garth Gayle, always does his best to make his country proud. ... |
| **Divide-Conquer Evaluator Re-Evaluation Reasons** | • This sentence is consistent with the article. The article states that Usain Bolt will compete at the IAAF/BTC World Relays in the Bahamas next month, which is the relay championship mentioned in the summary. The dates and his participation as part of the Jamaican team are also mentioned in the article
• This sentence is consistent with the article. The article mentions that Usain Bolt, a six-time Olympic gold medallist, will be part of the Jamaican team at the IAAF/BTC World Relays.
• This sentence is not consistent with the article. The article does mention that Bolt always does his best to make his country proud, but it is Bolt himself who says this, not the IAAF/BTC General Secretary Garth Gayle.'

... |
| **Second Round Improved Answer** | Usain bolt will compete at the relay championship on may 2 and 3 as part of the jamaican team. The six-time olympic gold medalist will be part of jamaica 's team at the iaaf/btc world. Bolt, as stated by himself, always does his best to make his country proud. ... |

## A.2 FULL SETS OF PROMPTS

Table 7: Semantic Consistency Divide-Conquer Evaluator Prompt

---

You are an evaluator. You will be given a true answer and an attempt answer.
True answer is the ground truth answer. Attempt answer is the answer you want to evaluate.

**Criteria:**
Your task is to evaluate whether the attempt answer is consistent with the true answer. You will evaluate it by:

* Listing all the aspects in the attempt answer
* Compare if each aspects exist in the true answer
* If it does, compare if the information in attempt answer is consistent with what is in the true answer
* It is OK that not all information from true answer exist in attempt answer

**Given:**
True Answer
{*true answer*}
Attempt Answer
{*answer to evaluate*}

**Task**
Work in a step by step way to make sure we get the right answer. You will format the output in json as follows:
{"reason": [{"sentence": "original sentence", "reason": "why this sentence is or is not consistent with the true answer"}], "is_consistent" : true/false}

Here is the evaluation in JSON format:

---

Table 8: Summarization Consistency Divide-Conquer Evaluator Prompt

You are an evaluator. You will be given an article and a summary.
Summary contains a summarized version of the article.

**Criteria:**
Your task is to evaluate whether the summary is consistent with the article. You will evaluate it by going through each sentence of the summary and check against the following procedures:

* Understands all the aspects in the sentence, who is doing what at when and where and what are the impact etc.
* Compare if each aspects exist in the article
* If it does, compare if the information in this sentence is consistent with what is in the article
* Compare if all the information in this sentence can be directly inferred or entailed from what is in the article, including but not limited to who, what, when, where etc.
* It is OK that not all information from article exist in this summary

**Given:**
Article
{*article*}
Summary
{*summary*}

**Task**
Work in a step by step way to make sure we get the right answer. You will format the output in json as follows:
{"reason": [{"sentence": "original sentence", "reason": "why this sentence is or is not consistent with the article. You should start with "this sentence is consistent with the article"or "this sentence is not consistent with the article"}], "is_consistent" : true/false}

Here is the evaluation in JSON format:

Table 9: Auto-Metric Converter Prompt

You are an evaluator. You will be given a list of paragraphs about "attempt answer". Your job is to:
* Identify whether each paragraph is positive or negative
* If the paragraph is positive, mark it as 1,
* If the paragraph is negative, mark it as -1.
* Output the mark for each paragraph in a json array

Example
**Given paragraphs:**
*"The attempt answer is incorrect as it states that employees in the US are not eligible to participate in the ESPP, which contradicts the true answer. So it is incorrect",
*"The attempt answer adds a new aspect that is not in the true answer.",
*"Yet it does list the correct article.And that is helpful."

**Thought:**
The first paragraph is negative as it mentions the attempt answer is wrong. Thus mark -1
The second paragraph is negative as it adds something that is not in true answer. Thus mark -1
The third paragraph is positive. Thus mark 1

**Answer:**
{"reason": ["The first paragraph is negative as it mentions the attempt answer is wrong. Thus mark -1", "The second paragraph is negative as it adds something that is not in true answer. Thus mark -1", "The third paragraph is positive. Thus mark +1"], "answer": [ -1, -1, 1]}

**Given:**
Attempt Answer:
{*attempt answer*}

**Answer:**

Table 10: Reason-Assisted Improver Prompt

You are a good writer. You will be given:
* An article
* A list of objects, each have two fields: sentence and reason
  ** sentence: These sentences are summaries for the given article.
  ** reason: These are the reasons why the sentence is consistent with the article or not.

Your job is to rewrite these sentences:
* If the sentence is consistent with the article, you can keep it as it is
* If the sentence is not consistent with the article, you can re-write it to make it consistent with the article based on the reasons given.

**Article**
{*article*}
**Sentences**
{*sentences*}

**Task**
Work in a step by step way to make sure we get the right answer. You will format the output in json as follows:
[ {"sentence": "original sentence", "improved_sentence": "improved sentence", "reason": "if it is improved, how it is improved. if not, say 'ALREADY CONSISTENT'"}]

Table 11: Paragraph Level Evaluator Prompt

---

You are an evaluator. You will be given an article and a summary.
Summary contains a summarized version of the article.

**Criteria:**
Your task is to evaluate whether the summary is consistent with the article. You will evaluate it by going through the summary and check against the following procedures:

* Understands all the aspects in the summary, who is doing what at when and where and what are the impact etc.
* Compare if each aspects exist in the article
* If it does, compare if the information in this summary is consistent with what is in the article
* Compare if all the information in this summary can be directly inferred or entailed from what is in the article, including but not limited to who, what, when, where etc.
* It is OK that not all information from article exist in this summary

**Given:**
Article
{*article*}
Summary
{*summary*}

**Task**
Work in a step by step way to make sure we get the right answer. You will format the output in json as follows:
{"reason": {"sentence": "original sentence", "reason": "why this summary is or is not consistent with the article. You should start with "this summary is consistent with the article"or "this summary is not consistent with the article"}, "is_consistent" : true/false}

Here is the evaluation in JSON format:

---

Table 12: Paragraph Level Reason Assisted Improver Prompt

---

You are a good writer. You will be given:
* an article
* a summary of the article
* a reason listing why the summary is or is not consistent with the article

Your job is to rewrite this summary:
* if the summary is consistent with the article, you can keep it as it is
* if the summary is not consistent with the article, you can rewrite it to make it consistent with the article based on the reasons given.

**Article**
{*article*}
**Summary**
{*summary*}
**Reason**
{*reason*}

**Task**
Work in a step by step way to make sure we get the right answer. You will format the output in json as follows:
{"sentence": "original summary", "improved_summary"": "improved summary"", "reason": "if it is improved, how it is improved. if not, say 'ALREADY CONSISTENT'"}

---

## A.3 DIVIDE-CONQUER EVALUATOR EXAMPLES

Table 13: QQP Evaluation Example

| True Answer | Attempt Answer | Is_Consistent | Reasons |
|---|---|---|---|
| How do I read and find my YouTube comments? | How can I see all my Youtube comments? | True | The sentence is asking the same question as the true answer, just phrased differently. Both are asking about how to view one's own comments on YouTube |
| What are the laws to change your status from a student visa to a green card in the US, how do they compare to the immigration laws in Canada? | What are the laws to change your status from a student visa to a green card in the US? How do they compare to the immigration laws in Japan? | False | • This sentence is consistent with the true answer as it asks about the laws to change status from a student visa to a green card in the US.
• This sentence is not consistent with the true answer. The true answer asks for a comparison with the immigration laws in Canada, not Japan." |
| If I fire a bullet backward from an aircraft going faster than the bullet; will the bullet be going backwards? | Do bullets travel faster than the speed of sound when shot from a gun? If not, is it possible? If they do, what gun and how much devastation occurs? | False | • This sentence is not consistent with the true answer because it is asking about the speed of bullets in relation to the speed of sound, not about the direction of a bullet when fired from a fast-moving aircraft.
• This sentence is not consistent with the true answer because it is asking about the possibility of bullets traveling faster than the speed of sound, not about the direction of a bullet when fired from a fast-moving aircraft.
• This sentence is not consistent with the true answer because it is asking about the type of gun and the amount of devastation, not about the direction of a bullet when fired from a fast-moving aircraft. |

Table 14: PAWS Evaluation Example

| True Answer | Attempt Answer | Is_Consistent | Reasons |
|---|---|---|---|
| What were the major effects of the cambodia earthquake , and how do these effects compare to the Iquique earthquake in 1877 ? | What were the major effects of the Iquique earthquake , and how do these effects compare to the cambodia earthquake in 1877 ? | False | The sentence is not consistent with the true answer because it reverses the order of the earthquakes. The true answer asks about the effects of the Cambodia earthquake and compares it to the Iquique earthquake in 1877, while the attempt answer asks about the effects of the Iquique earthquake and compares it to the Cambodia earthquake in 1877. |
| How do I start learning electronic music production ? Should I begin with books or videos ? I have logic pro 9 and I already made some songs but I would like to start from scratch ( as if I did not know anything ) . | How do I begin learning electronic music production ? Should I start with books or videos ? I have logic pro 9 and I already made some songs but I would like to start from scratch ( as if I did not know anything ) . | True | • This sentence is consistent with the true answer. The only difference is the use of the word 'begin' instead of 'start', but they mean the same thing.
• This sentence is consistent with the true answer. It is exactly the same.
• This sentence is consistent with the true answer. It is exactly the same. |

Table 15: SummEval Evaluation Example

| **Original Article** | Paul Merson has restarted his row with Andros Townsend after the Tottenham midfielder was brought on with only seven minutes remaining in his team 's 0-0 draw with Burnley on Sunday . 'Just been watching the game , did you miss the coach ?  RubberDub  7minutes , ' Merson put on Twitter . Merson initially angered Townsend for writing in his Sky Sports column that 'if Andros Townsend can get in ( the England team ) then it opens it up to anybody . ' Paul Merson had another dig at Andros Townsend after his appearance for Tottenham against Burnley Townsend was brought on in the 83rd minute for Tottenham as they drew 0-0 against Burnley Andros Townsend scores England 's equaliser in their 1-1 friendly draw with Italy in Turin on Tuesday night The former Arsenal man was proven wrong when Townsend hit a stunning equaliser for England against Italy and he duly admitted his mistake . 'It 's not as though I was watching hoping he would n't score for England , I 'm genuinely pleased for him and fair play to him ‚Äì it was a great goal , ' Merson said . 'It 's just a matter of opinion , and my opinion was that he got pulled off after half an hour at Manchester United in front of Roy Hodgson , so he should n't have been in the squad . 'When I 'm wrong , I hold my hands up . I do n't have a problem with doing that - I 'll always be the first to admit when I 'm wrong . ' Townsend hit back at Merson on Twitter after scoring for England against Italy Sky Sports pundit Merson ( centre ) criticised Townsend 's call-up to the England squad last week Townsend hit back at Merson after netting for England in Turin on Wednesday , saying 'Not bad for a player that should be 'nowhere near the squad ' ay @ PaulMerse ? ' Any bad feeling between the pair seemed to have passed but Merson was unable to resist having another dig at Townsend after Tottenham drew at Turf Moor . |
|---|---|
| **Original Summary Divided into Bullet Points** | • Paul merson was brought on with only seven minutes remaining in his team 's 0-0 draw with burnley. 
 • Andros townsend scored the tottenham midfielder in the 89th minute . 
 • Paul merson had another dig at andros townsend after his appearance . 
 • The midfielder had been brought on to the england squad last week . 
 • Click here for all the latest arsenal news news . |
| **Reasons** | • This sentence is not consistent with the article because it was Andros Townsend, not Paul Merson, who was brought on with only seven minutes remaining in the game. 
 • This sentence is not consistent with the article because it does not mention that Andros Townsend scored in the 89th minute. Also, the phrase 'scored the Tottenham midfielder' is not clear. 
 • This sentence is consistent with the article. The article mentions that Paul Merson had another dig at Andros Townsend after his appearance. 
 • This sentence is not consistent with the article because it does not mention that Andros Townsend was brought onto the England squad last week. 
 • This sentence is not consistent with the article because the article does not provide any latest Arsenal news. |
| **Is_Consistent** | False |

Table 16: QAGS-XSUM Evaluation Example

| | |
|---|---|
| **Original Article** | London 's first history day will be held on the anniversary of big ben 's first day in operation . It will be first celebrated on 31 may in 2017 with celebrations and events run by historic england . The date was decided upon after a poll involving 1,000 londoners . It was closely followed by 5 september - the date of the great fire of london . The yougov questionnaire also declared the houses of parliament as the building that best sums up london . People voted for the queen as their favourite historic london hero for the moment she secretly joined the crowds to celebrate victory in europe day . The results of the poll were released to mark the launch of historic england 's " keep it london " campaign . People were asked to select a date to celebrate the capital 's history , their historic hero and the building that sums up london . Big ben 's first day in operation was 31 may 1859 . The campaign is intended to encourage londoners to notice , celebrate and speak up for the heritage of their city , historic england said . The public body has also launched a film entitled i am london , which celebrates the historic buildings and places that have borne witness to the capital 's history . Duncan wilson , chief executive of historic england , said : " our heritage is a source of pride , a reminder of the city 's past , a foundation for its present and the building blocks for its future . |
| **Original Summary Divided into Bullet Points** | • Big ben 's 150th anniversary has been chosen as the date to celebrate london 's history . |
| **Reasons** | • This sentence is not consistent with the article. The article states that London's first history day will be held on the anniversary of Big Ben's first day in operation, which is 31 May 1859. However, it does not mention that this date is the 150th anniversary of Big Ben. |
| **Is_Consistent** | False |

Table 17: QAGS-CNN Evaluation Example

| | |
|---|---|
| **Original Article** | A southern iowa chiropractor accused of accepting sex as payment for his services and performing exorcisms on patients has surrendered his state license . The iowa board of chiropractic released a report wednesday detailing charges against charles manuel , of lamoni . Manuel signed an agreement last month admitting his misdeeds and pledging not to apply for reinstatement for at least 10 years . Patient satisfaction : a chiropractor in iowa has surrendered his license to practice and admitted to swapping services for sex and performing exorcisms on some patients . Nonetheless , he 's received outstanding evaluations for patient satisfaction on healthgrades . Com . The agreement requires that manuel prove any circumstances surrounding the revocation of his license no longer exist before he can resume practicing chiropractic in the state . Those circumstances included bartering sex for services with some patients . Manuel also recommended that patients stop taking medication he prescribed to them . A woman who answered a call to manuel 's home from the des moines register declined to comment on the case . A woman at his former practice said he had n't worked there for some time . A lamoni address listed on the yelp page for manuel 's practice appears to be a home on a residential street . While maneul has received just three patient survey responses on healthgrades . Com , those responses were quite positive . The disgraced chiropractor received a perfect five out of five stars in patient satisfaction . Strange practice : charles manuel , who admitted wrongdoing to the iowa board of chiropractic , listed his practice 's addresses on this residential street in the small agricultural town of lamoni in southern iowa . |
| **Original Summary Divided into Bullet Points** | • A chiropractor in iowa has surrendered his license to practice and admitted to swapping services for sex and performing exorcisms on some patients.
• Manuel also recommended that patients stop taking medication no longer exist before he can resume practicing chiropractic in the state .
• The disgraced chiropractor received a perfect five out of five stars in patient satisfaction . |
| **Reasons** | • This sentence is consistent with the article. The article states that Charles Manuel, a chiropractor in Iowa, surrendered his license and admitted to accepting sex as payment for his services and performing exorcisms on patients.
• This sentence is not consistent with the article. The article does mention that Manuel recommended patients stop taking medication he prescribed to them, but it does not state that this condition must be met before he can resume practicing.
• This sentence is consistent with the article. The article mentions that despite his misdeeds, Manuel received outstanding evaluations for patient satisfaction on healthgrades.com, including a perfect five out of five stars.'] |
| **Is_Consistent** | False |

### A.4 EXAMPLE OF SETTING $\alpha$ AND $\beta$ IN EQUATION 3

$\alpha$ and $\beta$ are added to ensure we are using a fully sentence level analysis. Below we give an example on how to use them to achieve this goal.

Suppose the output from Divide-Conquer Evaluator with customized prompt is:

```
1 {
2   "is_consistent" : False ,
3   "reasons" : [
4     "The two paragraphs are not consistent.",
5     "This sentence is consistent."
6     "This sentence is not consistent."
7   ]
8 }
```

After inspecting the *reasons* we notice that the first entry is not a sentence level analysis, but a paragraph level one. Thus when calculating scores we should remove the impact of this line.

Since in this case the first entry is negative, it will be given -1 score, thus we should set $\alpha = 1$ to mitigate this effect. Similarly, because we only have one entry that is not sentence level, we would set $\beta = -1$.

Thus for this particular case, equation 3 will be:

$$\mathcal{Z} = \left( \sum_{i=1}^{k} z_i + 1 \right) /(k-1), \quad \hat{\mathcal{Z}} = (\mathcal{Z} + 1)/2, \quad \hat{\mathcal{Z}} \in [0, 1] \tag{5}$$

## A.5 WHAT DOES LLM THINK OF SENTENCE LEVEL APPROACH?

To have some more fun, we asked different LLMs on what they think the best approach to evaluate consistency is, token level, paragraph level, or sentence by sentence level. We have a unanimous decision :)

Table 18: LLM's opinion on our sentence level approach

| Input Prompt | *Given a reference, we have a generated paragraph from LLM. Our objective is to check whether the generated text is consistent to the reference or not. Can you you tell me, to achieve the best accuracy, would you choose to compare them token-by-token, direct check of paragraphs as a whole, or sentence by sentence check on the semantic meaning?* |
|---|---|
| **GPT 3.5** | To achieve the best accuracy in checking the consistency of generated text with a reference, it is recommended to compare them sentence by sentence on the semantic meaning. This approach allows for a more nuanced analysis of the generated text and can identify inconsistencies that may not be apparent when comparing them token-by-token or as a whole paragraph. Additionally, comparing them sentence by sentence allows for a more efficient and targeted review of the generated text, as inconsistencies can be identified and addressed on a sentence-by-sentence basis. |
| **GPT 4** | To achieve the best accuracy, it would be most effective to check sentence by sentence on the semantic meaning. This is because language can be complex and nuanced, and a direct token-by-token or whole paragraph comparison might miss subtle differences in meaning. By checking sentence by sentence, you can ensure that the generated text accurately reflects the reference in both content and context. |
| **PaLM2** | To achieve the best accuracy in checking whether a generated paragraph from LLM is consistent with a reference, it would be best to compare them sentence by sentence on the semantic meaning. This is because comparing token-by-token would be too time-consuming and would not take into account the overall meaning of the sentences. Direct check of paragraphs as a whole would also not be as effective, as it would not identify specific inconsistencies between the sentences. Comparing them sentence by sentence on the semantic meaning would allow for a more thorough and accurate check of the consistency between the generated paragraph and the reference. |
| **Claude 2** | Overall, I would recommend the sentence-by-sentence semantic comparison approach. By checking the semantic consistency at a sentence level, it provides a good balance between catching inconsistencies in meaning while allowing for minor paraphrasing and word order changes. The paragraph-level comparison is too holistic while the token-by-token approach is too strict. Focusing on sentence-level meaning comparisons allows catching semantic inconsistencies with reasonably high accuracy. |

