# OpenReview forum: "Evaluating and Improving Generation Consistency of Large Language Models via A Divide-Conquer-Reasoning Approach"
_ICLR.cc/2024/Conference — Submitted to ICLR 2024_

### Official Review · Reviewer_Zcqt · 2023-10-27

**Soundness:** 1 poor
**Presentation:** 4 excellent
**Contribution:** 2 fair
**Rating:** 5
**Confidence:** 4

**Summary:**

Authors propose a three-step algorithm for detecting and improving the consistency between a reference piece of text and a generated piece of text. The algorithm begins by segmenting the input texts, then using an LLM to generate a justification statement for each generated sentence. In the next step, the justifications are used as input to another LLM to produce numerical predictions, to be aggregated to output a single score. The final step is to use a third LLM to re-write the generated sentences if they are predicted to be inconsistent.

The method is evaluated in two paraphrase detection and two summarization datasets, and shows a lot of improvement over BERTScore, BARTScore, and a few other metrics.

**Strengths:**

Very well-written paper, both in terms of writing and also in terms of presentation.\
Very good results.\
The research task is real, and has real world applications.

**Weaknesses:**

My argument is very simple: What if all the improvement that the authors are getting is coming from using GPT3.5/4 ? I am saying, it is not fair to compare GPT3.5/4 to BERT, or even to GPT3.\
Authors describe a long list of techniques and methods to justify their model. What if all of them are just distraction, and the main reason that their method works is because of good outputs generated by GPT3.5/4 ?\
In fact the authors report an experiment that supports my argument. Tables 2-4 compare the results of their method when GPT3.5 is replaced with GPT4. The improvements in most cases are substantial. Which means a lot of work is carried by the underlying LLM.\
In my opinion, all the experiments should be repeated with identical underlying LM or LLM to be able to claim that the method really works (please just do not say that because a method is called BERTScore, then we should only use BERT vectors in it, and for example BLOOM vectors cannot be used in it).

My second argument is to refute the authors argument about the robustness of their method against hallucination. On Page 2, the authors state that their model does not rely on LLMs, they also repeatedly claim that existing methods are prone to hallucination---multiple times in the intro section and in the other sections.\
 But they are using LLMs through out their algorithm! I don’t know how they cannot see that! Plus all the three steps of their algorithm can be easily subject to hallucination as well. For example in the first step the LLM can easily generate hallucinated justification. Why are the authors so certain that this cannot happen?! Not clear to me.

**Questions:**

None

---

> ### Author Response · Authors · 2023-11-18
> **Response to Reviewer Zcqt - part 1/2**
>
> Thank you very much for your detailed and thoughtful comments. We are glad that you found our paper well-written, our results strong, and the task practical. We address your questions and concerns below.
>
> >**W1: What if all the improvement that the authors are getting is coming from using GPT3.5/4 ? I am saying, it is not fair to compare GPT3.5/4 to BERT, or even to GPT3.**
>
> Our DCR method benefits from GPT3.5/4 but is not entirely dependent on GPT3.5/4 only. Our improvement is owing to our unique novel contributions of divide-conquer-reasoning strategies achieved by three LLM agents, DCE, AMC, and RAI. We leveraged the strength of GPT3.5/4 in our components but did not directly and completely rely on them for generating final outputs. Also, we highlight that unlike previous methods such as BertScore and G-Eval, DCR not only outputs a numeric metric but also clarifies the reasons behind the number. Such reasons enable humans to interpret and evaluate the DCR system and also empower the inconsistency-mitigation process, which is not available through BertScore or G-Eval.
>
>
> To verify our effectiveness, we focus on the comparison with recent state-of-the-art studies, such as G-Eval with GPT-3.5 and GPT-4 on multiple benchmarks. **To ensure a fair comparison, we paid special attention in our experiments to use identical underlying LLM as G-Eval used, i.e., GPT-3.5 and GPT-4 to report empirical results**. As Table 3 shows, our DCE-AMC-3.5 (0.592 and 0.563) outperforms G-Eval-3.5 (0.386 and 0.318) by a large margin. Similarly, our DCE-AMC-4 (0.700 and 0.668) also presents superior performance compared to G-Eval-4 (0.507 and 0.425) on SummEval datasets.
>
>
> We understand your concern about the effect of LLM on the final performance. Integrating multiple models, including BERT, BART, Llama, GPT-based, PaLM, Claude, etc with our DCR strategy is a promising and ambitious future research direction but is out of the scope of our current study. The current focus is GPT-3.5/4 and we have demonstrated strong advantages compared with G-Eval on multiple benchmarks.
>
>
> On Page 8, we provide a discussion about the effect of LLM models. We noted that DCE-AMC-4 generally outperforms DCE-AMC-3.5 across most datasets, which suggests that GPT-4 can further enhance performance, especially for more complex evaluation tasks. Nonetheless, the benefits of GPT-3.5, such as higher computational efficiency and lower API costs, should not be overlooked.

---

> ### Author Response · Authors · 2023-11-18
> **Response to Reviewer Zcqt - part 2/2**
>
> > **W2: But they are using LLMs through out their algorithm! I don’t know how they cannot see that! Plus all the three steps of their algorithm can be easily subject to hallucination as well. Why are the authors so certain that this cannot happen?!**
>
> *First*, our complete statement on Page 2 is that “Our approach does not rely on LLMs, which are prone to hallucination,  **to output numeric scores without justification**”. Our intention was to emphasize that the DCR framework does not rely on LLMs to **directly** generate scores, which is in contrast to previous methods such as G-Eval, which may be possibly hallucinated.  As we discussed in Section 2, existing LLM-based evaluators have a major drawback as the generated verbal scores by LLMs are prone to hallucinations, resulting in abnormally higher ratings for LLM-generated content that diverge from human judgment. Such methods also generate no actionable insight to justify the score or mitigate inconsistencies after identifying them.
>
> To overcome these issues, we intentionally avoid using LLM in the critical steps that involve generating numerical evaluation scores but instead use LLMs to perform reasoning which they have been proven to excel in. To operationalize this, we introduce an automatic metric converter (AMC) that aims to quantitatively measure the consistency evaluation by converting the reasons a numeric score system. AMC functions as a binary sentiment classifier that classifies the reasons to be either positive (marked by “+1" if the sentence is consistent), or negative (marked by “-1” otherwise) for each sentence, then utilizes this score array to calculate a comprehensive score to evaluate consistency, as explained in Section 3.2.  We believe such a numerical score calculated by AMC is more stable than the verbal score directly output by LLMs, as is verified by experiments.
>
> We clarify this sentence in the revised version as “**Our approach does not rely on LLMs to directly output numeric scores without justification, which are prone to hallucination**”.
>
> ---
>
> *Second*, we clarify that we never state that LLM hallucinations cannot happen in our framework. On the contrary, we acknowledge hallucination as a key impediment to broader adoptions of LLMs in many critical domains which motivates our research that aims to design an automated framework for assessing inconsistency of LLM-generated contents to facilitate effective hallucination mitigation and reduction. This is through acknowledging that although currently there is no fundamental fix to LLM hallucinations, with the recent advances in generative AI there are certain tasks that LLM can achieve superhuman performance and are less prone to hallucination, such as summarization task [1].  Therefore, what we pursue is to leverage the strength of LLM in downstream consistency evaluation and improvement to minimize the risk and negative effect of LLM in open-domain generation tasks. Specifically, we take the following innovative approaches to mitigate LLM hallucinations in our framework,
> - Unlike direct paragraph check via LLM, our divide-conquer strategy (via DCE component) focuses on sentence level, which defines a more specific task for LLM to check consistency.
> - Rather than directly generating verbal scores by LLM, we use LLMs to generate reasons and translate these reasons into numeric scores (via ACE component) by calculating mathematically.
> - Instead of ending after consistency evaluation, we propose a multi-round improvement strategy (via RAI component), which eliminates the probability of LLM hallucinations occurring in a specific round, and also provides a more consistent response eventually.
>
> Beyond the above three strategies, we also carefully verify our approach and results via
>
> - Manual examination of the output response. There are some illustrative examples in Table 6, 13-17.
> - Experiments on evaluating the overall accuracy of the framework on three benchmarks, see Figure 3. Below is a table of the same results.
>
> While LLM hallucination possibly happened in each component, our DCR framework (DCE+AMC+RAI) demonstrates high accuracy and robust performance in consistency evaluation and improvement on all benchmarks.
>
> | Metric  Name  | F1 | Precision  | Recall |
> | --------  | --------  |--------  |--------  |
> | SUMMEVAL | 0.933 | 0.929 | 0.936|
> | QAGS-XSUM | 0.811 | 0.718 | 0.930|
> | QAGS-CNN| 0.871 | 0.842 | 0.901|
>
> [1] Pu, Xiao, Mingqi Gao, and Xiaojun Wan. "Summarization is (almost) dead." arXiv preprint arXiv:2309.09558 (2023).

---

> > ### Author Response · Authors · 2023-11-20
> > **Follow-up on Rebuttal**
> >
> > Dear Reviewer Zcqt,
> >
> > We sincerely appreciate the insightful feedback you provided on our paper. As we are approaching the end of the discussion phase, we would like to take a moment to ensure that we have adequately addressed your concerns and answer any remaining questions.
> >
> > Could you kindly confirm if our responses have satisfactorily addressed all your questions? Are there any other questions or concerns we could address to merit a higher score? We are more than willing to make the necessary revisions to enhance the quality of our submission if there are any additional concerns.
> >
> > We greatly appreciate the time and effort you have dedicated to reviewing our work and eagerly await your further guidance.
> >
> > Authors

---

> > > ### Author Response · Authors · 2023-11-21
> > > **Further Discussions and Feedback Welcome**
> > >
> > > We deeply appreciate the insights you've shared during the review process. Following our revisions and previous responses, we are genuinely curious if we have adequately addressed the concerns you raised. We believe we have done.
> > >
> > > We would appreciate it if you could kindly let us know if there are any further questions. In the extremely limited time remaining, we are still eager to do our utmost to address them!

---

> ### Comment · Reviewer_Zcqt · 2023-11-22
>
> My apologies for the slow response---like yourselves I am a researcher too, and always struggling with my schedule.
>
> Thank you for the clarifications.
>
> I am not convinced with your explanations about the lack of comparison with other methods. Yes, I agree with you, having a fair setup between all the metrics to achieve a realistic conclusion is ambitious. But so are your claims in the paper, in comparison to what other metrics do. In my opinion, the method is not a significant scientific step. In other words, I am not learning anything new from your method, which is breaking down paragraphs and individually processing them, and then asking another model to generate the scores for them. It would be significant, if you showed that out of all the metrics, if they use exactly the same tools, none of them achieve these results, which meant none of them perform these simple operations (breaking down and generating scores). I personally doubt this is the case, and you did not resolve my doubts. Unfortunately, you just reiterated your arguments in the paper.
>
> Regarding the second issue, which is your arguments about hallucination and the use of LLMs. I am not convinced that it was just one sentence, otherwise i would not written this in the review:
> > ... they also repeatedly claim that existing methods are prone to hallucination---multiple times in the intro section and in the other sections ....
>
> but I am willing to revise my score. Because it is just a simple issue and is fixable, and now that you are aware of it, you can resolve it and improve your paper. The current review rating is "Reject", I wish I could increase the rating just by 0.5 or by 1, but it looks the only option is to increase if by 2!
>
> ***
>
> ps. please take this with a grain of salt, but when i was reading the experimental section of your paper, i was thinking that this is like comparing a lamborghini to a minivan. And arguing that the reason that the lamborghini can drive faster than the minivan is because of its aerodynamic design! but we both know that this is not true. the reason is the powerful engine inside the lamborghini.\
> all the evidence that i am seeing supports the same conclusion about your method. it appears that the main performance improvement is coming from the engine (GPT3.5/4 models).

---

> ### Author Response · Authors · 2023-11-23
> **Response to Reviewer Zcqt**
>
> We understand the tight schedule and appreciate your taking the time to respond before the due date. The following is our response to your comment *sentence-by-sentence*.
>
>
> >*“Yes, I agree with you, having a fair setup between all the metrics to achieve a realistic conclusion is ambitious. But so are your claims in the paper, in comparison to what other metrics do.”*
>
> We note that our contribution is not merely to design a metric for evaluation but also to develop a method for automatically improving the consistency of the generated text without requiring human supervision.
>
> >*“… which is breaking down paragraphs and individually processing them, and then asking another model to generate the scores for them.”*
>
> To clarify, **we don't rely on generated scores from any model**. As previously highlighted, verbal scores generated by LLMs are prone to hallucinations, which is a well-documented issue that has been acknowledged in prior research and verified in our experiments.
>
> >*“Regarding the second issue, which is your arguments about hallucination and the use of LLMs. I am not convinced that it was just one sentence, otherwise i would not written this in the review”*
>
> We have meticulously examined that the term “hallucination” appeared 21 times in our paper (14 times in the main paper and 7 times in the references), primarily used in context with “mitigation” and “detection”. The sentence in the first paragraph of page two is the only single occurrence where we see that might have caused confusion. If the reviewer believes that there exist other cases where we might have misstated our contribution, we kindly ask you to point us to it such that we can revise it in our next version.
>
> >*“but I am willing to revise my score. Because it is just a simple issue and is fixable, and now that you are aware of it, you can resolve it and improve your paper. The current review rating is "Reject", I wish I could increase the rating just by 0.5 or by 1, but it looks the only option is to increase if by 2!”*
>
> Your acknowledgment of this being an addressable issue and your willingness to reevaluate the score are greatly appreciated.
>
> >*“… but when i was reading the experimental section of your paper, i was thinking that this is like comparing a lamborghini to a minivan. And arguing that the reason that the lamborghini can drive faster than the minivan is because of its aerodynamic design! but we both know that this is not true. the reason is the powerful engine inside the lamborghini … it appears that the main performance improvement is coming from the engine (GPT3.5/4 models).”*
>
> We respectfully disagree with this argument. We have indeed compared with other LLM-powered baselines such as G-eval and GPT-Score, which are also based on GPT series. **In particular, our experimental results summarized in Table 3 & Table 4 show that our method consistently outperforms G-eval across all considered metrics, and the comparison is done under the same GPT version, i.e., DCE-AMC-3.5 outperforms G-eval-3.5 and DCE-AMC-4 outperforms G-eval-4**. Note that G-eval and GPT-Score are both state-of-the-art works on NLG evaluation (published in 2023) and are well-cited (cited 141 and 105 times respectively according to Google Scholar).
>
> Finally, we would like to stress our contribution in leveraging the latest advancements in generative models to craft a framework that optimizes LLMs’ capabilities while circumventing known pitfalls, which is a direction aligned with numerous recent studies [1-11]. We believe such research is a pivotal step towards the practical deployment of LLMs.

---

> ### Author Response · Authors · 2023-11-23
> **References**
>
> **Recent studies**
>
> [1] Liu, Yang, Dan Iter, Yichong Xu, Shuohang Wang, Ruochen Xu, and Chenguang Zhu. "Gpteval: Nlg evaluation using gpt-4 with better human alignment." arXiv preprint arXiv:2303.16634 (2023). EMNLP 2023.
>
> [2] Fu, Jinlan, See-Kiong Ng, Zhengbao Jiang, and Pengfei Liu. "Gptscore: Evaluate as you desire." arXiv preprint arXiv:2302.04166 (2023).
>
> [3] Manakul, Potsawee, Adian Liusie, and Mark JF Gales. "Selfcheckgpt: Zero-resource black-box hallucination detection for generative large language models." arXiv preprint arXiv:2303.08896 (2023). EMNLP 2023.
>
> [4] Wang, Jiaan, Yunlong Liang, Fandong Meng, Haoxiang Shi, Zhixu Li, Jinan Xu, Jianfeng Qu, and Jie Zhou. "Is chatgpt a good nlg evaluator? a preliminary study." arXiv preprint arXiv:2303.04048 (2023).
>
> [5] Zeng, Zhiyuan, Jiatong Yu, Tianyu Gao, Yu Meng, Tanya Goyal, and Danqi Chen. "Evaluating large language models at evaluating instruction following." arXiv preprint arXiv:2310.07641 (2023).
>
> [6] Liu, Minqian, Ying Shen, Zhiyang Xu, Yixin Cao, Eunah Cho, Vaibhav Kumar, Reza Ghanadan, and Lifu Huang. "X-Eval: Generalizable Multi-aspect Text Evaluation via Augmented Instruction Tuning with Auxiliary Evaluation Aspects." arXiv preprint arXiv:2311.08788 (2023).
>
> [7] Madaan, Aman, Niket Tandon, Prakhar Gupta, Skyler Hallinan, Luyu Gao, Sarah Wiegreffe, Uri Alon et al. "Self-refine: Iterative refinement with self-feedback." arXiv preprint arXiv:2303.17651 (2023). NeurIPS 2023.
>
> [8] Liu, Yuxuan, Tianchi Yang, Shaohan Huang, Zihan Zhang, Haizhen Huang, Furu Wei, Weiwei Deng, Feng Sun, and Qi Zhang. "Calibrating LLM-Based Evaluator." arXiv preprint arXiv:2309.13308 (2023).
>
> [9] Chan, Chi-Min, Weize Chen, Yusheng Su, Jianxuan Yu, Wei Xue, Shanghang Zhang, Jie Fu, and Zhiyuan Liu. "Chateval: Towards better llm-based evaluators through multi-agent debate." arXiv preprint arXiv:2308.07201 (2023).
>
> [10] Gao. Mingqi, Jie Ruan, Renliang Sun, Xunjian Yin, Shiping Yang, and Xiaojun Wan. "Human-like summarization evaluation with chatgpt." arXiv preprint arXiv:2304.02554 (2023).
>
> [11] Jason, Wei, Xuezhi Wang, Dale Schuurmans, Maarten Bosma, Brian Ichter Fei Xia, Ed H. Chi, Quoc V. Le, Denny Zhou “Chain-of-Thought Prompting Elicits Reasoning in Large Language Modelsz”. arXiv:2201.11903 (2022) NeurIPS 2022.

---

> > ### Comment · Reviewer_Zcqt · 2023-12-01
> >
> > I don't understand what technique the authors are using in discussing scientific subjects. Whatever it is, it is very confusing. Specifically for reviewers, whom unfortunately, are given the privilege to be impatient. I suggest that the authors instead of arguing with reviewers, try to understand what the reviewers are complaining about and directly address those.
> >
> > **
> >
> > The authors in their latest comment say: **we don't rely on generated scores from any model**.\
> > This is not true, in Section 3.2 of the paper they use +1 and -1 scores to obtain an overall score. Are not these called scores? Then what are they?!
> >
> > The authors asked me to give them another example of confusing descriptions in their paper (other than what I already gave them in the main review), here is another one: In the "Limitation of Existing Methods", on Page 3, they say:
> > > these approaches have a major drawback as the generated verbal scores by LLMs are prone to hallucinations, resulting in abnormally higher ratings for LLM-generated content that diverge from human judgment
> >
> > I quoted the entire sentence, so that, I do not get accused of cherry picking from their text again.\
> > So now the question is this: are not you yourself using LLMs to generate scores? Yes you are. Then why are you claiming that this a weakness that other methods have?
> >
> > I understand that authors are unable to enter into another round of discussion with me (!), but I am increasing my score to marginally below acceptance threshold

---

### Official Review · Reviewer_9wcP · 2023-10-31

**Soundness:** 3 good
**Presentation:** 3 good
**Contribution:** 2 fair
**Rating:** 3
**Confidence:** 4

**Summary:**

This paper proposes a sentence-granularity evaluation strategy based on LLM.
The evaluation is decomposed into pre-sentence assessments, which are then combined into a single score.
The experiments show that sentence-level evaluation is more consistent than token-level (e.g., BertScore) or passage-level assessment (e.g., GPT-Score) for paraphrasing and summarization tasks.

**Strengths:**

1 Assessing the quality of candidate answers is best done using fine-grained signals such as sentence-level granularity.

2 The proposed method achieves higher consistency with human annotations in tasks of paraphrasing identification and summarization, compared to a wide range of baseline methods.

**Weaknesses:**

1 The proposed method has a limited scope and is not easily generalizable to broader instruction-following tasks. In the experiment, the method required hand-crafted prompts for each specific task, which can be overwhelming and even infeasible when evaluating a wide spectrum of tasks. Other LLM-based evaluators can handle this common scenario more effectively.

2 The experiments lack human evaluations beyond the instance level, leaving unclear the overall accuracy of sentence-level judgments.

3 The per-sentence assessment protocol is used to identify and fix inconsistencies, but it may be prone to LLM's overconfidence -- it prefers its predictions even if it is wrong.

4 The evaluation focuses solely on consistency, but other criteria like factuality and coherence are also important for text generation. It's unclear why only consistency was considered.

**Questions:**

1 Why it is stated in the Intro that DCE “does not rely on LLM“, considering Chatgpt/GPT4 is required for most components of the method?

2 How is the reason-assisted improver (RAI) compared to self-refine?

3 G-Eval does not use references during evaluation, so comparing the proposed method to G-Eval may not be fair since they use different evaluation approaches.

---

> ### Author Response · Authors · 2023-11-18
> **Response to Reviewer 9wcP - part 1/3**
>
> Thank you for your detailed review and suggestions to improve the paper. We are glad to hear that you found that our idea is good for assessing the quality of candidate answers, and our method achieves better performance than a wide range of baselines. We address your concerns and questions below.
>
> >**W1: The proposed method has a limited scope and is not easily generalizable to broader instruction-following tasks. In the experiment, the method required hand-crafted prompts for each specific task, which can be overwhelming and even infeasible when evaluating a wide spectrum of tasks. Other LLM-based evaluators can handle this common scenario more effectively.**
>
> Our work focuses on the evaluation and improvement of generated text consistencies. We do not claim DCR as a generic evaluation metric and our experiments demonstrated that DCR achieves the state-of-the-art performance on consistency evaluation compared with the existing baseline methods. Our DCR is easily generalizable to assess the consistency between any two text sequences without constraints on their format, that is, they can be either generated texts by LLM or written by human experts.
>
> We agree that our DCR framework requires hand-craft prompts for specific tasks, and acknowledge that this is a general hurdle shared by all works relying on LLMs, which include G-Eval, GPTScore, and self-refine.  Specifically, in G-Eval, different prompts will need to be composed for different aspects: consistency, coherence, etc.  Self-refine defines multiple customized prompts to perform their INIT - FEEDBACK – REFINE components. Our current solution is to structure our prompts in a modularized manner so task-specific content can be updated easily. An automated prompt-tuning procedure is beyond the focus of our study. If you believe there exists coinciding work on an LLM-based evaluator that offers a more effective solution, we kindly ask you to point us to it.
>
> >**W2-3: The experiments lack human evaluations beyond the instance level, leaving unclear the overall accuracy of sentence-level judgments. The per-sentence assessment protocol may be prone to LLM's overconfidence -- it prefers its predictions even if it is wrong.**
>
> Although the experiments lack human evaluations on sentence-level judgments, our DCR method provides an overall numerical score by calculating and integrating sentence-level scores through an automatic metric converter (AMC) component. As shown in Fig.3 and the following table result, we verify the F1-score, precision, and recall performance of our method on sentence-level and paragraph-level evaluations compared with human evaluations. Such high accuracy provides solid support for our reason-assisted improver (RAI) which aims at reducing inconsistencies via multi-round iterations. We agree with the reviewer that LLMs have the potential tendency to be overconfidence. However, as demonstrated by our experiments, such overconfidence can be mitigated through the careful design of our framework, resulting in high accuracy that is well-aligned with human intuition.
>
> | Metric  Name  | F1 | Precision  | Recall |
> | --------  | --------  |--------  |--------  |
> | SUMMEVAL | 0.933 | 0.929 | 0.936|
> | QAGS-XSUM | 0.811 | 0.718 | 0.930|
> | QAGS-CNN| 0.871 | 0.842 | 0.901|

---

> ### Author Response · Authors · 2023-11-18
> **Response to Reviewer 9wcP - part 2/3**
>
> >**W4: The evaluation focuses solely on consistency. It's unclear why only consistency was considered.**
>
> Assessing the consistency of LLM-generated content is crucial for ensuring AI safety and has become a critical step in improving the reliability of many real-world application domains that rely on LLM by preventing the generation of misinformation and harmful content. For example, consistency checking can significantly enhance the chain of thought reasoning in LLMs [1]. Recent studies employ consistency checking to detect hallucinations [2] based on pre-trained LLMs and instruction tuned LLMs [3]. Although these methods exhibit promising results on several specific tasks, including mathematical reasoning and factual assessment, the potential failures of self-consistency are often overlooked [4].
>
> This is essentially due to a lack of a generic, automatic, and reliable strategy that assesses the consistency of two responses, let alone remediating such inconsistency after identifying them. We highlight that our objective is not to propose a generic or comprehensive evaluation pipeline to handle broader metrics as G-Eval or GPTScore claimed. Instead, we mainly focus on the consistency scope and have achieved our state-of-the-art performance across multiple benchmarks in semantic, factual, and summarization consistency tasks, compared to existing baseline methods. Furthermore, our approach also substantially reduces nearly 90\% output inconsistencies, showing promise for effective hallucination mitigation and reduction.
>
> [1] Xuezhi Wang, Jason Wei, Dale Schuurmans, Quoc Le, Ed Chi, Sharan Narang, Aakanksha Chowdhery, and Denny Zhou. Self-consistency improves chain of thought reasoning in language models. arXiv preprint arXiv:2203.11171, 2022.
>
> [2] Potsawee Manakul, Adian Liusie, and Mark JF Gales. Selfcheckgpt: Zero-resource black-box hallucination detection for generative large language models. arXiv preprint arXiv:2303.08896, 2023.
>
> [3] Niels Mündler, Jingxuan He, Slobodan Jenko, and Martin Vechev. Self-contradictory hallucinations of large language models: Evaluation, detection and mitigation. arXiv preprint arXiv:2305.15852, 2023.
>
> [4] Angelica Chen, Jason Phang, Alicia Parrish, Vishakh Padmakumar, Chen Zhao, Samuel R Bowman, and Kyunghyun Cho. Two failures of self-consistency in the multi-step reasoning of llms. arXiv preprint arXiv:2305.14279, 2023.
>
> >**Q1: Why it is stated in the Intro that DCE “does not rely on LLM“**
>
> Our complete statement on Page 2 is that “Our approach does not rely on LLMs, which are prone to hallucination,  **to output numeric scores without justification**”. Our intention was to emphasize that the DCR framework does not rely on LLMs to **directly** generate scores, which is in contrast to previous methods such as G-Eval, which may be possibly hallucinated.  As we discussed in Section 2, existing LLM-based evaluators have a major drawback as the generated verbal scores by LLMs are prone to hallucinations, resulting in abnormally higher ratings for LLM-generated content that diverge from human judgment. Such methods also generate no actionable insight to justify the score or mitigate inconsistencies after identifying them.
>
> To overcome these issues, we intentionally avoid using LLM in the critical steps that involve generating numerical evaluation scores but instead use LLMs to perform reasoning which they have been proven to excel in. To operationalize this, we introduce an automatic metric converter (AMC) that aims to quantitatively measure the consistency evaluation by converting the reasons a numeric score system. AMC functions as a binary sentiment classifier that classifies the reasons to be either positive (marked by “+1" if the sentence is consistent), or negative (marked by “-1” otherwise) for each sentence, then utilizes this score array to calculate a comprehensive score to evaluate consistency, as explained in Section 3.2.  We believe such a numerical score calculated by AMC is more stable than the verbal score directly output by LLMs, as is verified by experiments.
>
> We clarify this sentence in the revised version as “**Our approach does not rely on LLMs to directly output numeric scores without justification, which are prone to hallucination**”.

---

> ### Author Response · Authors · 2023-11-18
> **Response to Reviewer 9wcP - part 3/3**
>
> >**Q2: How is the reason-assisted improver (RAI) compared to self-refine?**
>
> Thanks for the suggestion.  As self-refine contains the process of INIT - FEEDBACK - REFINE, we summarize the difference between DCR with self-refine as follows.
>
> - DCR aims at achieving a similar goal as self-refine in providing supervision-free refinements to LLM-generated content. The focus of the two frameworks is different as self-refine aims at more general purposes with the ability to customize prompts for individual tasks whereas DCR aims at detecting and mitigating inconsistency.
>
> - There is no specific prompt dedicated to consistency checking within the context of self-refine, where it is simply defined as a criterion of "Consistent" in one specific task, i.e., dialogue-response. Also, self-refine uses 6 in-context examples for feedback generation, but the DCR is a zero-shot approach that does not require few-shot examples.
>
> - Comparing RAI to the REFINE step in self-refine, both approaches use LLM as the improver. Thus, it boils down to the feedback input and the prompt for performance differences. As the FEEDBACK prompt is more general regarding dialogue in self-refine, the feedback given from DCE is more specific.
>
> >**Q3: G-Eval does not use references during evaluation, so comparing the proposed method to G-Eval may not be fair since they use different evaluation approaches.**
>
> We apologize for the confusion caused by the definition of “reference” in our DCR framework. We would like to make the following clarifications:
>
> 1. Our DCR method employs the same setting as G-Eval, which does not rely on a golden reference written by the human expert or ground truth labels.
>
> 2. Our DCR method aims to offer a generic way to assess the consistency between two text sequences. For instance, in the summarization task, one text sequence may be the original document, while the other might be the generated text summarization by LLMs. To distinguish between the two text sequences, in our paper we referred to one of the text sequences as “reference” and the other as “candidate”. Accordingly, we formulate the problem to check whether the candidate is consistent with the reference or not, as presented in Section 2.  We realize that such wording might have caused confusion and will change them to more neutral terms such as “text sequence 1” and “text sequence 2”.
>
> Below we provide a detailed explanation of the “reference” used in our experiments:
> - **Paraphrase detection tasks**. For example, in the Quora Question Pairs (QQP) dataset, each question pair is annotated with a binary value indicating whether the two questions are paraphrases of each other.   We consider “question1” as the “reference” and “question2” as the “candidate”, and our task is to evaluate if the candidate is consistent with the reference in semantic meaning.
> - **Summarization tasks**. For example, SummEval datasets include original source articles, machine summaries, and human summaries. Our “reference” in this task is the original source article, and our “candidate” is the machine summaries. Our task is to check the factual consistency between them without relying on any additional golden reference or ground truth.

---

> > ### Author Response · Authors · 2023-11-20
> > **Follow-up on Rebuttal**
> >
> > Dear Reviewer 9wcP,
> >
> > We sincerely appreciate the insightful feedback you provided on our paper. As we are approaching the end of the discussion phase, we would like to take a moment to ensure that we have adequately addressed your concerns and answer any remaining questions.
> >
> > Could you kindly confirm if our responses have satisfactorily addressed all your questions? Are there any other questions or concerns we could address to merit a higher score? We are more than willing to make the necessary revisions to enhance the quality of our submission if there are any additional concerns.
> >
> > We greatly appreciate the time and effort you have dedicated to reviewing our work and eagerly await your further guidance.
> >
> > Authors

---

> > > ### Author Response · Authors · 2023-11-21
> > > **Further Discussions and Feedback Welcome**
> > >
> > > We deeply appreciate the insights you've shared during the review process. Following our revisions and previous responses, we are genuinely curious if we have adequately addressed the concerns you raised. We believe we have done.
> > >
> > > We would appreciate it if you could kindly let us know if there are any further questions. In the *extremely limited time* remaining, we are still eager to do our utmost to address them!

---

### Official Review · Reviewer_2NAw · 2023-10-31

**Soundness:** 3 good
**Presentation:** 3 good
**Contribution:** 2 fair
**Rating:** 6
**Confidence:** 3

**Summary:**

This paper investigates consistency evaluation by employing a Large Language Model (LLM) as an evaluation agent. The authors introduce a "divide-and-conquer" prompting technique to break down both candidate and reference documents into individual sentences. This approach facilitates comparison at the sentence level initially, before combining these sentence-level results to produce a final evaluation. The proposed methodology is tested on SUMMeval consistency and QAGS benchmarks. Results indicate an improvement in correlation with human evaluators, specifically on the consistency criterion.

**Strengths:**

- This paper introduces an efficient "divide-and-conquer" evaluation prompt designed to enhance the assessment of the consistency criterion through an LLM agent.
- The manuscript is both well-structured and easy to understand.

**Weaknesses:**

- Novelty and Contribution:
  - The approach of SMART [1], which suggests using sentences as the basic units for text evaluation, seems similar to the divide-and-conquer technique introduced in this paper. Specifically, methods like SMART-Bleurt and SMART-CHRF have shown strong agreement with human evaluations. Given that, the novelty of the divide-and-conquer strategy that focuses on sentence-level evaluation might not be very original when considering what SMART has already introduced. This makes me question the overall unique contribution of the present work.

[1] SMART: Sentences as Basic Units for Text Evaluation

- Evaluation Fairness:
  - The authors use a "reference" document to help with consistency evaluation. However, from the prompts shown in Appendix A.2, it's unclear what "reference" means. In some cases, it seems to mean the "true answer" (like a gold standard in common situations), while in others, it refers to the "article" (or source document in common contexts). Could the authors clarify what they mean by "reference"?

  - G-EVAL is described as a method that doesn't rely on a reference (reference-free method), where "reference" seems to mean a gold reference written by the human expert. If this study's prompts use the "gold reference" or the "source", it would be helpful if the authors could specify whether the proposed divide-and-conquer technique is based on a reference or not, and if it uses a source or not.

  - Additionally, the lack of comparisons with other consistency methods like SMART, NLI, and Bleurt makes me question the thoroughness of the paper.

Update after author feedback:

The authors have addressed my queries regarding evaluation fairness and have conducted additional experiments comparing with the "consistency method." Given the constraints of the rebuttal period and the quality of open source code, they were able to partially reproduce the results from the SMART paper. This effort has largely resolved my concerns about evaluation fairness.

Consequently, I am increasing my overall recommendation score to 6 (weak accept).

However, I maintain that the novelty of the "divide and conquer" approach is somewhat limited, as the SMART paper has already proposed using "sentence as evaluation unit." Additionally, the paper could benefit from revisions for methodological clarity.

**Questions:**

Given that the reference and candidate documents might have varying numbers of sentences, how do you handle sentence-level comparisons? Specifically, do you employ any sentence matching techniques, and if so, how are they implemented?

---

> ### Author Response · Authors · 2023-11-18
> **Response to Reviewer 2NAw - part 1/2**
>
> Dear reviewer 2NAw,
>
> Thank you very much for the constructive comments and feedback. We are happy to hear that the reviewer found our method efficient, and our draft is both well-structured and easy to understand. Below, we have tried to address all of your feedback and questions. Please take a look and let us know in case you would like additional clarification on any of these points.
>
> > **W1: Could the authors clarify what they mean by "reference"? It would be helpful if the authors could specify whether the proposed divide-and-conquer technique is based on a reference or not, and if it uses a source or not.**
>
> 1. Our DCR method employs the same setting as G-Eval, which does not rely on a golden reference written by the human expert or ground truth labels.
>
> 2. Our DCR method aims to offer a generic way to assess the consistency between two text sequences. For instance, in the summarization task, one text sequence may be the original document, while the other might be the generated text summarization by LLMs. To distinguish between the two text sequences, in our paper we referred to one of the text sequences as “reference” and the other as “candidate”. Accordingly, we formulate the problem to check whether the candidate is consistent with the reference or not, as presented in Section 2.  We realize that such wording might have caused confusion and will change them to more neutral terms such as “text sequence 1” and “text sequence 2”.
>
> Below we provide a detailed explanation of the “reference” used in our experiments:
> - **Paraphrase detection tasks**. For example, in the Quora Question Pairs (QQP) dataset, each question pair is annotated with a binary value indicating whether the two questions are paraphrases of each other.   We consider “question1” as the “reference” and “question2” as the “candidate”, and our task is to evaluate if the candidate is consistent with the reference in semantic meaning.
> - **Summarization tasks**. For example, SummEval datasets include original source articles, machine summaries, and human summaries. Our “reference” in this task is the original source article, and our “candidate” is the machine summaries. Our task is to check the factual consistency between them without relying on any additional golden reference or ground truth.
>
> > **W2: The lack of comparisons with other consistency methods like SMART, NLI, and Bleurt makes me question the thoroughness of the paper.**
>
> Thanks for your suggestion. We have added some additional experiments for comparison. Since SMART has not released open-source code yet, we cannot evaluate its performance in our experiment setting, i.e., the summary-level correlation on SummEval datasets. Instead, we implement the system-level correlation used in SMART, for a fair comparison of these baseline methods. As shown in the table below, our method outperforms SMART in terms of all metrics, including the SMART series provided in their paper.
>
> | Metric   | S1-CHRF| S2- CHRF | SL- CHRF | S1-BLEURT| S2-BLEURT | SL-BLEURT | DCE-AMC (ours) |
> | -----------------  | ------- |------- |------- |------- |------- |------- |------- |
> | Kendall tau | 0.733 | 0.700 | 0.733 | 0.667 | 0.750 | 0.567 | **0.799** |

---

> ### Author Response · Authors · 2023-11-18
> **Response to Reviewer 2NAw - part 2/2**
>
> > **W3: Given that the reference and candidate documents might have varying numbers of sentences, how do you handle sentence-level comparisons? Specifically, do you employ any sentence matching techniques, and if so, how are they implemented?**
>
> We would like to clarify that the sentence-level comparison strategy used in our DCR framework is not to compare each sentence in the candidate text sequence to each sentence from the reference text sequence (sentence-to-sentence), but to compare each sentence in the candidate text sequence to the entire reference text sequence (sentence-to-paragraph). This design reduces the number of comparison operations and ensures semantic consistency. For instance, in the SummEval task, our objective is to check if the candidate summaries are consistent with the original source article. This is achieved by checking if each sentence in the candidate summaries is consistent with the whole original articles, rather than a sentence-by-sentence matching. Thus, our method does not rely on any sentence-matching techniques, such that there is no issue with the varying number of sentences.
>
> > **W4: Compare with SMART paper and our unique contribution.**
>
> There are several key differences we would like to highlight:
>
> - SMART leverages the reference summaries for the SUMMEVAL database, which relies on both "reference summaries" and "original articles" for checking. SMART is therefore a reference-based method. However, our DCR method is a reference-free method, as explained above, without requiring any additional references.
>
> - Our DCR framework does not rely on any sentence-matching techniques.  Instead, our task is to check if each sentence in the candidate is consistent with the original article, which is longer in size and more comprehensive, thus rendering the consistency checking more difficult in our opinion. We overcome this challenge by leveraging LLM agents and show outperformed results.
>
> - Our DCR framework is not only an evaluation metric but also provides an effective solution to consistency improvement. This is achieved by introducing the reasoning component. Our reason-assisted improver, implemented by an LLM agent, allow human to understand the logic and build the foundation for inconsistency mitigation.

---

> > ### Comment · Reviewer_2NAw · 2023-11-18
> >
> > Thank you for the response! Here is the code link to SMART https://github.com/google-research/google-research/tree/master/smart_eval

---

> > > ### Author Response · Authors · 2023-11-19
> > > **Our additional experiments and thanks for sharing the code link**
> > >
> > > Dear reviewer 2NAw,
> > >
> > > Thanks very much for sharing the code link!
> > >
> > > We were not able to run the code as is due to changes in its package. So we made changes to
> > > - [this line|
> > > https://github.com/google-research/google-research/blob/513d75625c30a9080d8afdcc1ba1bde46c573d62/smart_eval/matching_functions.py#L89]
> > > to be `return [
> > >           sacrebleu.sentence_chrf(c, [r]).score for r, c in zip(reference, candidate)].
> > >
> > > We also did something similar to BleuMatchingFunction
> > > - [ |https://github.com/google-research/google-research/blob/513d75625c30a9080d8afdcc1ba1bde46c573d62/smart_eval/matching_functions.py#L104] as well.
> > >
> > > For sanity check, after such changes, we are able to replicate the CHRF results in Table 1 in the paper.
> > >
> > > However, the `src_can_pairs.bleurt.csv` and `ref_can_pairs.bleurt.csv` files downloaded from the command `gsutil cp -r gs://gresearch/SMART ./` as instructed do not have enough entries when we try to replicate the BLEURT series results. Thus we treat the score as 1.0 for such cases in
> > > -  [here|https://github.com/google-research/google-research/blob/513d75625c30a9080d8afdcc1ba1bde46c573d62/smart_eval/summeval_utils.py#L177C13-L177C13]
> > > and
> > > - [here|https://github.com/google-research/google-research/blob/513d75625c30a9080d8afdcc1ba1bde46c573d62/smart_eval/summeval_utils.py#L187 ].
> > >
> > > With such changes, the system-level highest BLEURT  score we calculate is 0.6, different from the 0.75 in the paper.
> > >
> > > With the changes mentioned, the performance comparison is shown in the table below:
> > >
> > > |               | S1-CHRF	 | S2- CHRF | SL- CHRF | S1-BLEURT  | S2-BLEURT  | SL-BLEURT  | DCE-AMC (ours) |
> > > |---------------|---------|----------|----------|------------|------------|------------|----------------|
> > > | System-level  | 0.733   | 0.700    | 0.733    | 0.667      | 0.750      | 0.567      | **0.799**        |
> > > | Summary-level | 0.306   | 0.328    | 0.294    | 0.110      | 0.114      | 0.109      | **0.668**        |
> > >
> > > Let us know if you have any further questions.

---

> > > > ### Comment · Reviewer_2NAw · 2023-11-20
> > > >
> > > > Dear Authors,
> > > >
> > > > Thank you for your efforts in expanding the experiment. Although the current results are limited by time constraints and open resource quality, I am confident these issues will be addressed in your next version. Regarding the challenges faced with SMART, I recommend reaching out directly to the first author of SMART for assistance.
> > > >
> > > > Based on these considerations, I am adjusting my score to 6 (weak accept), indicating my support for including this work in the conference.

---

> > > > > ### Author Response · Authors · 2023-11-20
> > > > > **Thanks for you positive comments - Reviewer 2NAw**
> > > > >
> > > > > Dear Reviewer 2NAw,
> > > > >
> > > > > We would like to express our sincere appreciation for your constructive feedback and thoughtful reviews that helped to improve our paper and your support in favor of accepting our paper. We will reach out to the first author of SMART to fix the open-source issues and include a comprehensive comparison in our final version.
> > > > >
> > > > > Authors

---

### Author Response · Authors · 2023-11-18
**Summary of Author Response**

Dear reviewers,

Thank you all for taking the time to review our paper and we sincerely appreciate all the feedback. In particular, we feel encouraged to see that the reviewers find that

- **Our idea and research task are real and applicable**:  “The research task is real, and has real-world applications.” (reviewer Zcqt)
- **Our method is efficient and effective**:“ This paper introduces an efficient "divide-and-conquer" evaluation…” (reviewer 2NAw); “Assessing the quality of candidate answers is best done” (reviewer 9wcP)
- **Our experiments achieve strong performance**: “The proposed method achieves higher consistency with human annotations in tasks of paraphrasing identification and summarization, compared to a wide range of baseline methods.” (reviewer 9wcP); “Very good results.” (reviewer Zcqt)
- **Our paper is well-written and easy to understand**: “The manuscript is both well-structured and easy to understand” (reviewer 2NAw); “Very well-written paper, both in terms of writing and also in terms of presentation.” (reviewer Zcqt)

---

Based on the reviewer's feedback, we will make the following changes to further improve the clarity of the manuscript:

- Revised corresponding text to clarify that our DCR method is a reference-free method, which does not rely on a golden reference written by the human expert or ground truth labels. This setting is the same as G-Eval.
- Added a further explanation of our definition of reference in our experiments.
- Added a clarification of the relationship between our DCR method and LLMs. What we emphasize is that the DCR framework does not rely on LLMs to directly generate scores for consistency evaluation, which may be possibly hallucinated.
- Added additional experiments to compare DCR with SMART and highlight our unique strength and novel contribution.
- Added a clarification that our DCR is to check whether the candidate summaries are consistent with the original source article rather than a sentence-by-sentence matching.
- Added a clarification of our scope on consistency and our performance in handling LLM’s overconfidence.
- Added a detailed description of the difference between our DCR and self-refine, and highlighted our advantages.
- Added a detailed explanation about how three components in DCR, specifically the AMC component,  address LLM hallucinations and mitigate inconsistency, as well as our verification via manual examination and experiments.

We are committed to address all comments and we welcome any further questions or discussions.

Authors of paper 6094

---

### Meta-Review · Area_Chair_EfTg · 2023-12-12

**Metareview:**

The paper aims to evaluate the consistency of LLM-generated texts using a divide-and-conquer strategy. The proposed method first develops a divide-and-conquer evaluator (DCE) to break down the comparison between two generated responses into individual sentences, and then introduces an automatic metric converter (AMC) that translates the output from DCE into a numeric score. After that, it leverages the analytical reasons with explanations identified by DCE to generate new responses to reduce any inconsistencies. The reviewers generally applauds the paper's idea of conducting evaluation on the sentence level and the achieved good results. They also engaged in discussions during rebuttal in general. However, the reviewers still have concerns after the discussion period: (1) lack of novelty, as sentence-level evaluation has been explored below. (2) claims in the paper being very confusing or misleading, e.g., the one related to hallucination. Given all the reviews and discussions, the paper would benefit from another round of revision.

**Justification For Why Not Higher Score:**

Please see the above summary.

**Justification For Why Not Lower Score:**

N/A

---

### Decision · Program_Chairs · 2024-01-16

Reject